# Measuring Agents in Production

**Melissa Z. Pan**[1][*]  **Negar Arabzadeh**[1][*]  **Riccardo Cogo**[2]  **Yuxuan Zhu**[3]  **Alexander Xiong**[1]  **Lakshya A Agrawal**[1]
**Huanzhi Mao**[1]  **Emma Shen**[1]  **Sid Pallerla**[1]  **Liana Patel**[4]  **Shu Liu**[1]  **Tianneng Shi**[1]  **Xiaoyuan Liu**[1]
**Jared Quincy Davis**[4]  **Emmanuele Lacavalla**[2]  **Alessandro Basile**[2]  **Shuyi Yang**[2]  **Paul Castro**[5]  **Daniel Kang**[3]
**Koushik Sen**[1]  **Dawn Song**[1]  **Joseph E. Gonzalez**[1]  **Ion Stoica**[1]  **Matei Zaharia**[1][*]  **Marquita Ellis**[5][*]

[1]UC Berkeley   [2]Intesa Sanpaolo   [3]UIUC   [4]Stanford University   [5]IBM Research

## Abstract

LLM-based agents already operate in production across many industries, yet we lack an understanding of what technical methods make deployments successful. We present the first systematic study of **M**easuring **A**gents in **P**roduction, MAP, using first-hand data from agent developers. We conducted 20 case studies via in-depth interviews and surveyed 86 deployed systems practitioners across 26 domains. We investigate why organizations build agents, how they build them, how they evaluate them, and their top development challenges. Our study finds that production agents are built using simple, controllable approaches: 68% execute at most 10 steps before human intervention, 70% rely on prompting off-the-shelf models instead of weight tuning, and 74% depend primarily on human evaluation. Reliability (consistent correct behavior over time) remains the top development challenge, which practitioners currently address through systems-level design. MAP documents the current state of production agents, providing the research community with visibility into deployment realities and underexplored research avenues.

## 1. Introduction

Large language models enable a new class of software systems—**agents**—that combine foundation models with tools, memory, and reasoning to autonomously execute multi-step tasks (Wang et al., 2024b). LLM-based agents

---
*Project Co-Leads  [1]University of California at Berkeley  [2]Intesa Sanpaolo  [3]University of Illinois at Urbana-Champaign  [4]Stanford University  [5]IBM Research. Correspondence to: Melissa Z. Pan, Negar Arabzadeh, Marquita Ellis <{melissapan,negara,mme}@berkeley.edu>.

*Proceedings of the $43^{rd}$ International Conference on Machine Learning*, Seoul, South Korea. PMLR 306, 2026. Copyright 2026 by the author(s).

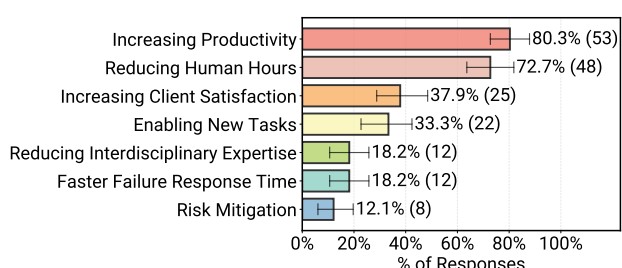

*Figure 1.* Reasons practitioners build and deploy AI agents ($N$=66). The question is multi-select, so proportions do not sum to 1. Error bars indicate 95th percentile intervals estimated from 1,000 bootstrap samples with replacement.

have gained substantial research interest, demonstrating potential in areas such as drug discovery and scientific discovery (Huang et al., 2025; Novikov et al., 2025; Lu et al., 2024). Industry is also now deploying agents in domains central to society: finance, healthcare, and education (IACPM, McKinsey, 2025; UHS, Inc., 2025; Linsenmayer, 2025).

Despite widespread excitement about agents, studies show agent deployments often fail or underdeliver (Xue et al., 2025; Reuters Staff, 2025; Shome et al., 2026). The stark contrast between the potential of agents and their failures raises the fundamental question of what enables successful agent deployment. The field can only advance collectively through shared understanding of real-world challenges and lessons learned. Yet, unfortunately, little information is publicly available on how production agents are built.

To address this knowledge gap, we present MAP, the first large-scale systematic study of AI agents in production. We study the practices of developers and teams behind successful real-world systems via four research questions (*RQ*s):

*RQ1.* What are the applications of agents?
*RQ2.* What models, architectures, and methods are used?
*RQ3.* How are agents evaluated?
*RQ4.* What are the top challenges in deployment?

We answer these questions by conducting 20 in-depth interviews with deployment teams ranging from major tech-

nology companies with AI research labs to agent startups (Figure 15), and surveying 306 practitioners actively building agents across 26 domains (Figure 2). We conducted the study from April to November of 2025. We filtered survey responses to 86 systems in production or pilot phases (Figure 14b), serving hundreds to millions of daily users (Figure 14c). We refer to these production and pilot systems as *deployed agents* and focus our analysis on them in the main paper. Full unfiltered survey data appear in §D. Our key findings for each RQ are the following:

*RQ1*: **Productivity gains drive agent adoption.** Practitioners deploy agents primarily to increase productivity (80%), mainly serving human users rather than other systems. Deployed agents operate in latency-tolerant applications: 66% allow response times of minutes or longer, as agents outperform human baselines even with these latencies.

*RQ2*: **Production agents favor simplicity and control.** Teams use off-the-shelf models rather than weight tuning (70%), rely on manual prompt construction (79%), and use static workflows (68% execute $\leq 10$ steps before human intervention). Organizations deliberately trade capability for controllability to maintain reliability.

*RQ3*: **Human-in-the-loop evaluation dominates.** Deployed agents rely primarily on human-in-the-loop evaluation (74%); other methods (e.g., LLM-as-a-judge) serve as complementary verification. Due to limited available benchmarks and challenges in creating them, 75% of teams forgo formal benchmarking, relying instead on A/B testing or expert feedback to improve reliability.

*RQ4*: **Reliability is the top development challenge.** Other critical challenges include evaluation (limited by benchmark scarcity and delayed feedback) and security (mitigated via operational constraints).

Our findings suggest an underlying principle: practitioners achieve reliability through best practices in system-level design rather than model-level or algorithmic advances. For example, despite the popularity of RL in research and its benchmark gains, practitioners default to prompting closed-source models because this approach is more robust to model upgrades and more sample-efficient. Thus, teams deliberately choose simple, controllable methods not from lack of sophistication, but because they offer reliable agent performance and fast development cycles.

MAP documents real-world agent practices with first-hand data from practitioners. By sharing deployment data, which are typically kept proprietary, we connect research advances with real-world constraints and challenges. We hope these data-driven insights can help inspire the community to address the underexplored research directions and technical challenges revealed by our study, advancing the field of agents together to deliver value in the real-world. The con-

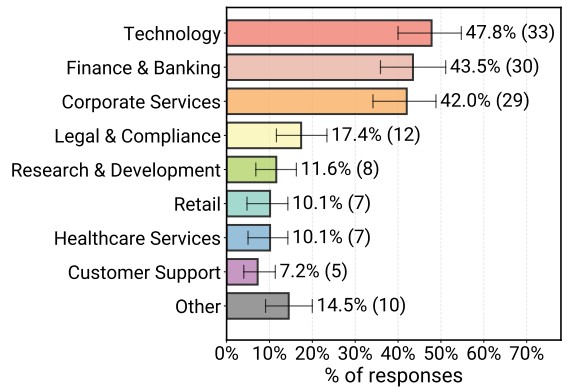

*Figure 2.* Application domains where practitioners deploy Agents ($N = 69$). ("other") domains are listed in Table 1. The question is multi-select i.e., a system may be assigned to multiple categories.

tributions of this paper are as follows:

1. **First large-scale empirical study of production agents:** We conduct 20 in-depth interviews with deployment teams and survey 306 practitioners, including 86 in deployments, providing systematic data on agents serving real users.
2. **Characterization across 17 design dimensions:** We provide quantitative and qualitative data on technical decisions, including models, architectures, prompting, evaluation, applications, and operational constraints.
3. **Data-driven insights:** We show practitioners achieve reliability through system-level design rather than algorithmic advances. Our data provides foundational evidence to support diverse research directions in agent systems.

## 2. Related Work

To our knowledge, we offer the first technical characterization of how agents running in production are built. Additional related work in each category appears in §F.

**Commercial Agent Surveys.** Prior work examines agent adoption from adjacent perspectives. Challapally et al. (2025) study economic viability from business executives' views at companies attempting agent integration. Shome et al. (2026) analyze marketing materials and conduct user studies, revealing capability gaps. LangChain (2024) surveyed 1,300 professionals on motivations and challenges. And various consulting reports focus on organizational readiness (PwC, 2025; Sukharevsky et al., 2025). Our work differs in scope (production systems) and focus (engineering-level technical data from practitioners).

**Research Literature Surveys.** Academic surveys synthesize published research, providing overviews of agent architectures and taxonomies (Wang et al., 2024b; Liu et al., 2025), with specialized surveys on evaluation (Yehudai et al., 2025) and security (He et al., 2025). In contrast, our empiri-

cal study collects primary data directly from practitioners building deployed agents through original interviews and surveys rather than synthesizing existing literature.

**Single-System Studies.** Companies publish on individual systems (Gottweis et al., 2025; Anthropic, 2024; Jackson et al., 2025) and open-source implementations (OpenAI, 2025a; OpenHands, 2025), providing valuable insights into specific systems. We report common patterns across diverse deployments rather than documenting individual systems.

## 3. Study Methodology

To understand the technical methods enabling the production of agents, we collected data through 20 in-depth interviews and surveyed 306 practitioners, conducting the study from April to November of 2025. However, such a study presents distinct challenges: (1) successful agent deployments with real-world users are scarce, and (2) production agents are typically proprietary, limiting company participation.

We address these challenges through systematic sampling and rigorous study design. Throughout the study, we prioritize diversity (domains and organization sizes) and transparency (documenting selection criteria and limitations). This study is exempted by our institutions' review board.

### 3.1. Data Collection #1: In-depth Case Studies

We conducted 20 case studies through interviews with ML and software engineers.

**Participants.** Rather than selecting all participants upfront, we iteratively expanded our sample guided by emerging findings following snowball sampling (Naderifar et al., 2017). We began with agent teams from major technology companies with significant AI research within our professional network, then systematically broadened across *application diversity*, *organizational maturity*, and *global reach*. This yielded cases spanning five sectors (Table 3), organization sizes from startups to global enterprises (Figure 15), and deployment scales from hundreds to millions of users across multiple continents. At the case level, 11 sources span five to six continents, 4 span two to four, and 5 operate within a single continent; by country reach, 13 span tens of countries, 1 spans hundreds, and 5 operate within one country (Figures 17 and 16, §C.2). All cases represent verified production (14) or pilot (6) deployments serving real users. We anonymized all case studies and refer to each as C01, C02, etc. in result sections (Table 3 in §B).

**Semi-structured interviews.** We conducted 30-90 minute interviews with teams of 2-5 researchers with no affiliation to the participating organizations following a protocol covering system architecture, evaluation, and deployment challenges (§C.3). The semi-structured format allowed us to explore unexpected themes while maintaining consistency, and each interview is documented under the same protocol.

**Analysis.** After each interview, researchers documented observed patterns for theme extraction in addition to interview notes. We then applied grounded theory's (Glaser & Strauss, 1967) open coding across all interviews to identify recurring themes, followed by focused coding to organize themes into broader categories. All open coding is performed by humans: at least three researchers independently code each interview note, and we resolve disagreements through peer debriefing.

### 3.2. Data Collection #2: Public Online Survey

**Survey design.** We developed a 47-question survey to confirm our qualitative observations from our case studies and to understand agents deployment at scale and address our four research questions. We piloted the survey with agent development teams outside our study, iteratively refining questions based on feedback. We implemented the final survey with dynamic branching in Qualtrics, where subsequent questions adapt based on prior responses, allowing relevant deep-dives while maintaining completion rates. All questions were optional to respect participants' freedom to answer. Complete survey questions appear in §G.

**Distribution and collection.** We distributed the validated survey through the Berkeley RDI Agentic AI Summit (UC Berkeley RDI, 2025), the AI Alliance Agents-in-Production Meetup (The AI Alliance, 2025), the UC Berkeley Agentic AI MOOC (Berkeley RDI, 2025), and professional networks (LinkedIn, Discord, X) from July 28 to October 29, 2025, targeting practitioners who build agent systems. We received 306 valid responses from practitioners across diverse technical roles (Figure 14a) and 26 application domains (Figure 2).

**Data filtering and processing.** We filtered responses to 86 data points explicitly in production or pilot phases. All results in §4–7 refer to these as deployed agents; full unfiltered data appear in §D. These deployed systems are not small pilots; many serve real users at scale, from tens to over one million (§4.3, Figure 14c). Because access to production agents is itself a major bottleneck, 86 deployed systems and 20 in-depth cases provide meaningful coverage. Most survey questions use structured formats (single-select, multi-select, numeric). For domain classification (Figure 2), the only free-text question, we applied LOTUS (Patel et al., 2025) for semantic aggregation to derive candidate domain categories and employed three independent human annotators for labeling (Cohen's $\kappa$=0.636); details appear in §B.1.2. For data on categorical comparisons, we report 95% confidence intervals computed using 1,000 bootstrap samples with replacement.

### 3.3. Study Limitations

Despite best efforts to obtain broad sample coverage, our study has inherent limitations in scope and sampling. Geographically, case study teams are concentrated in the Americas, with a few in Europe. Participation bias affects both data sources: agent teams accepted interview invitations based on company policies, while survey respondents likely skew toward our professional networks despite multi-channel distribution. Multi-month data collection broadened system coverage but may also have introduced temporal bias, which we mitigated in part by emphasizing themes that remained consistent over time. Our study captures agent deployments during a window from April to November 2025, in a rapidly evolving field. While some fine-grained patterns may shift over time, we expect the underlying principles and practical considerations identified in our study to remain useful as qualitative evidence of how practitioners approach agent deployment, rather than as fixed prevalence estimates for this limited period. Thus, we view MAP as an initiative towards documenting the leading edge of production agents practice for open research rather than a comprehensive coverage of all agents development globally. Interviews and surveys are well suited to our questions—why teams choose bounded workflows, how they evaluate model changes, and which constraints shape deployment—for which practitioner evidence is especially valuable. Direct auditing of deployed systems offers a complementary lens, which we are actively pursuing as follow-up work.

## 4. *RQ1 Results*: What Are the Applications of Agents?

We present findings from our study on why organizations build agents, which applications reach deployment, who uses them, and what requirements shape their design.

### 4.1. Motivation for Building Agents

Deployed agents primarily target measurable productivity gains. Among surveyed practitioners with deployed agents, 80% cite increased productivity, and 72% cite reduced human task-hours (Figure 1). Benefits that are harder to quantify emerge less frequently in current deployments, such as risk mitigation (12%) and reducing the need for interdisciplinary expertise (18%). Productivity-focused applications offer straightforward success metrics: interviewed teams commonly measure productivity gains by comparing total time to completion between agents and alternative systems. In contrast, operational improvements like risk mitigation require extended validation periods before benefits become measurable. Among practitioners who evaluated alternative systems for the same objectives, 83% prefer agents over non-agentic solutions (software or human execution).

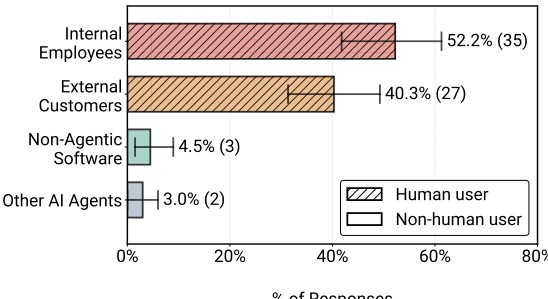

*Figure 3.* Distribution of primary end users for deployed agentic systems ($N$=67). Hatched bars (////) denote systems primarily serving human end users, while solid bars denote systems serving non-human end users (e.g., internal services or automated agents).

> **Finding 1:** Practitioners primarily build agents for productivity gains through automation, while harder-to-quantify uses like risk mitigation are less common.

### 4.2. Application Domains

Among 69 surveyed deployed agents, we observe 26 distinct application domains, extending well beyond software engineering. Figure 2 shows Finance & Banking (44%), Technology (48%), and Corporate Services (42%) lead, signaling early adoption. A substantial long tail follows across retail, healthcare and other service industries. This growing diversity suggests that many real-world tasks beyond traditional benchmarks (e.g., coding, mathematical reasoning) are viable candidates for agent applications. We believe the breadth of production agents signals opportunities and demand for research to advance agents for diverse contexts.

> **Finding 2:** Finance, technology, and corporate services lead agent adoption, but emerging deployments across 26 domains suggest expanding opportunities for agent applications in diverse real-world contexts.

### 4.3. Users of Agents

The deployed agents surveyed primarily serve human users: 92.5% target humans rather than other software systems (Figure 3). Internal employees comprise 52% of users, external customers 40%, and non-human systems only 8%. Case study interviews reveal this human-centric design may be deliberate. Interviewed practitioners report deploying internally first to mitigate reliability and security risks, where agent errors have lower consequences and human oversight is more available. Even external-facing systems typically augment domain experts rather than replacing human workers, with humans serving as final verifiers of agent outputs. Among surveyed systems, deployment scale varies: 43% serve hundreds of users while 26% serve tens of thousands to over 1 million daily users (Figure 14c).

**Finding 3:** 93% of surveyed deployed agents serve human users, enabling direct human oversight.

## 4.4. Latency Requirements

Survey data shows production agents tolerate surprisingly relaxed latency: 66% allow response times of minutes or longer, and 17% set no explicit limit (Figure 4). This pattern challenges mainstream optimization goals in machine learning systems research focused on latency reduction.

Interview data reveals the tolerance stems from a dominant use case: background automation of human workflows. Interviewed practitioners report that their minutes-scale agents still outperform human baselines by 10x, which is critical when staffing shortages exist and the automated tasks are secondary to human users' core job responsibilities. For example, `C01` deploys agents to automate clinicians' preparation and obtaining of insurance approval, `C02` assists sales personnel with customer care, and `C16` helps software engineers triage incidents. Fifteen of 20 case studies can operate asynchronously; some even batch process requests hourly or overnight. For these applications, minute-scale latency beats the alternative human completion time.

Only 5 of 20 cases require real-time responsiveness. These include voice agents operating at human conversation speeds (`C04-05`), where latency becomes the primary deployment challenge (Section B.4.2). For the majority, relaxed latency requirements allow practitioners to prioritize output quality and reliability over latency optimization. This pattern suggests opportunities for agents and techniques that trade speed for correctness and downstream performance.

**Finding 4:** Deployed agents tolerate minute-scale latency, with asynchronous systems emerging as popular agent applications.

## 5. *RQ2 Results:* What Models, Architectures, And Techniques Are Used?

We now examine five critical design decisions: model selection, model weights tuning methods, prompt construction, agent architectures, and development frameworks.

### 5.1. Model Selection

Case study data shows deployed agents rely heavily on proprietary frontier models. Seventeen of 20 case studies use closed-source models (Figure 5), with 10 teams explicitly reporting Anthropic Claude (Sonnet 4, Opus 4.1) or OpenAI GPT (o3), the state-of-the-art at interview time. Open-source adoption (3/20 cases) addresses specific constraints, such as high-volume workloads where inference costs are prohibitive (`C09` fine-tuned a model for infrastruc-

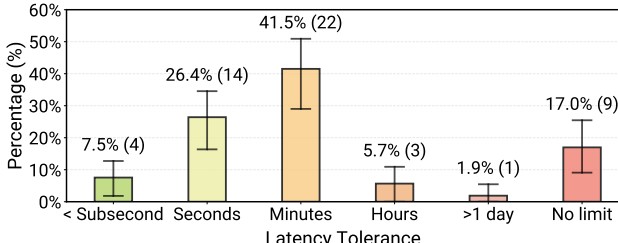

*Figure 4.* Reported tolerable end-to-end response latency for deployed agentic systems ($N$=53).

ture maintenance using existing company GPUs) or regulatory requirements preventing sensitive data sharing.

Model selection is based on empirical testing. Engineers evaluate the most powerful accessible models and selecting based on downstream performance. These teams report that runtime costs remain negligible compared to alternative expert labor costs (e.g., medical professionals, senior engineers), justifying the use of expensive frontier models.

**Finding 5:** Case study data shows deployed agents rely on proprietary frontier models (17/20); open-source addresses cost or regulatory constraints.

**Number of Distinct Models.** Survey data shows that while 41% of deployed agents use a single model, 59% coordinate multiple models. Case study interviews reveal that half of the teams (10/20) combine models driven by both *functional needs* and surprisingly *operational constraints*.

*Functional needs* include (1) cost optimization, routing simple subtasks to smaller models and complex reasoning to powerful models (`C16`), and (2) modality requirements, combining text-to-speech with LLMs (`C04-C05`) or pairing domain-specific models (chemistry) with general reasoning models (`C06-C08`, Figure 13).

*Operational constraints.* Interview data reveals some teams maintain multiple models to manage fragility from model upgrades. Agent scaffolds, prompts, and evaluations lock onto specific model behaviors. Deploying newer models can break agent workflows, forcing teams to run legacy models alongside updates (`C10`). This pattern reveals a reliability challenge: in production settings, newer or more capable models do not guarantee improved agent performance.

**Finding 6:** Use of multiple models in production agents (59%) reflects both functional needs (cost, modality) with operational constraints (model migration).

### 5.2. Model Weight Tuning

Case study data shows deployed agents overwhelmingly favor prompting over weight tuning. Fourteen of 20 cases (70%) use off-the-shelf models without supervised fine-

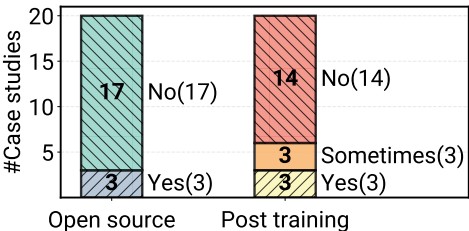

*Figure 5.* Distribution of model characteristics across case studies ($N$=20) for model source openness and post-training usage. "Sometimes" indicates selective use (e.g., when cost-justified).

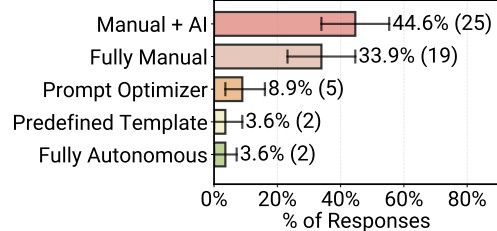

*Figure 6.* Distribution of system-prompt construction strategies across deployed Agentic AI systems ($N$=53).

tuning (SFT) or reinforcement learning (RL) (Figure 5). Two teams explicitly report foundation models already meet their task requirements, making post-training unnecessary.

Five of 20 cases use SFT (Figure 5): two teams apply SFT consistently, targeting business-specific contexts where domain knowledge improves performance (e.g., C17), while three apply it selectively for enterprise clients where training data is available and cost trade-offs make sense (e.g., C14). Even among these five cases, four combine fine-tuned models with off-the-shelf LLMs rather than relying on fine-tuning alone. Only one case uses RL-trained models (C06 for scientific discovery). Three teams report plans for future RL adoption in complex multi-system environments.

Interview data reveals practical considerations that favor prompting over post-training. Teams report that SFT and RL require substantial implementation effort and are brittle to model upgrades, requiring costly retraining when versions change. Interviewed teams prefer methods that reduce development and maintenance overhead.

Latency requirements can also potentially push teams toward model adaptation. Among the 4 cases with explicit bounded-latency requirements, 3 use weight tuning (C02, C04, C14) and 1 pairs a proprietary model with system-level techniques such as caching (C05). The sample is small, so we report this as an observation rather than a prevalence claim: latency-sensitive teams adapt the model or system design rather than relying solely on frontier proprietary APIs.

> **Finding 7:** Post-training is less common in deployments (6/20 cases). Interviewed teams rely primarily on prompt engineering with frontier models.

### 5.3. Prompt Tuning Strategies

Survey data shows humans remain central to prompt construction despite emerging automated methods. Among surveyed deployed agents, 34% use manual hard-coded prompts, and 45% use manual drafting augmented by LLMs (Figure 6). Only 9% use prompt optimizers (e.g., DSPy (Khattab et al., 2024)). Case study interviews con-

firm this pattern: only 1 of 20 teams explored automated optimization, while the rest rely on human construction, sometimes with LLM refinement. Interviewed practitioners report prioritizing controllable and interpretable methods enabling fast iteration, whereas we speculate black-box optimizations may incur additional engineering overhead.

Survey data shows prompt complexity varies widely with system maturity. While most deployed agents (52%) use short prompts under 500 tokens, some use very long prompts exceeding 10,000 tokens (Figure 9b). Case study interviews reveal that long prompts typically occur in external client-facing agents requiring extensive guardrails.

> **Finding 8:** Survey data shows humans dominate prompt construction (79% manual or manual+LLM); automated optimization remains rare (9%).

### 5.4. Agent Architecture

Case study data shows predefined structured workflows dominate production deployments. Eighty percent (16/20) of case studies use structured workflows rather than open-ended autonomous planning. These agents operate within scoped action spaces where practitioners define task sequences upfront. Nine cases implement sophisticated agentic RAG pipelines, single agents retrieving via tool calls, or pipelines with 20+ subtasks explicitly configuring retrieval at each step. For example, C01 follows a fixed sequence of coverage lookup, medical necessity review, and risk identification, where the agent autonomously completes each subtask but high-level objectives remain fixed. This pattern reflects practitioners adapting existing business processes into agent workflow rather than building fully autonomous AI workers. Interview data reveal that workflows dominate production to prioritize controllability and human-expert-in-the-loop oversight for reliability. Teams deliberately constrain autonomy for production stability. Only 1 case uses unconstrained exploration, exclusively in sandboxed environments with rigorous CI/CD verification (C09).

**Number of Steps Before Human Intervention.** Survey data quantifies this constrained autonomy: 68% of deployed agents execute fewer than 10 steps before requiring human intervention, and 47% execute fewer than 5 steps (Fig-

ure 7a). For comparison, research prototypes show substantially higher step counts (Figure 21a), reflecting aggressive autonomy exploration that consolidates during deployment. Interview data shows problem complexity, planning non-determinism, and latency drive intentional step limits.

> **Finding 9:** Both study data show structured workflows dominate; agents execute <10 steps (68%) before human intervention, prioritizing reliability over autonomy.

### 5.5. Agent Frameworks

As agent systems mature and scale, teams prefer custom agent implementations over third-party frameworks. Among case studies with production deployments, 85% (17/20) build custom in-house implementations with direct API calls; only 3 use external frameworks (LangChain/LangGraph, DSPy). Survey data shows 61% currently use frameworks (led by LangChain/LangGraph at 25%), but 2 teams explicitly migrated from frameworks (e.g., CrewAI) to custom solutions for production deployment.

Why custom over frameworks? Three drivers emerge. First, *flexibility*: production agents require vertical integration with proprietary infrastructure and data pipelines that rigid frameworks cannot support (`C14`: bespoke orchestration for varied client environments). Second, *simplicity*: core agent loops are straightforward to implement directly, avoiding dependency bloat (`C12`: custom ReAct implementation). Third, *security*: enterprise policies prohibit certain external libraries, forcing compliant internal solutions.

> **Finding 10:** Production agents favor custom implementations (85%) over frameworks for flexibility, simplicity, and control at scale.

## 6. *RQ3 Results:* How Are Agents Evaluated For Deployment?

We examine how practitioners assess agents for production readiness and runtime correctness, including what teams compare against and verification methods.

### 6.1. Baselines and Benchmarks

Survey data shows most teams lack standardized evaluation frameworks during agents development. Only 39% compare their agents systems to non-agentic baselines (existing software, human execution), while 26% report no meaningful baseline exists (Figure 10a). Interview data reveals that even when baselines exist, comparison proves challenging, where non-agentic systems often combine multiple components (human procedures, legacy software, document systems), making isolated technical comparison difficult (e.g., `C17`: HR agent replacing hybrid human-software workflow).

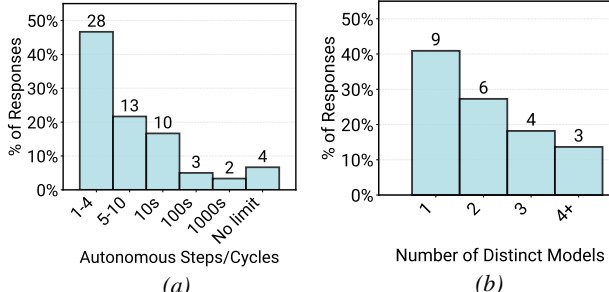

*Figure 7.* Agent architecture and model selection of deployed agents. (a) Number of autonomous execution steps before user intervention ($N$=60); (b) Number of models used in deployed systems ($N = 22$).

Case study data shows formal benchmarks are even rarer: 75% evaluate without benchmark sets, relying instead on A/B testing, user feedback, and production monitoring. We refer to benchmarks as curated task sets with expected outputs for systematic evaluation. Five teams build custom benchmarks, one from scratch with expert-labeled ground truth, four by synthesizing existing data (test cases, logs, support tickets). These benchmarks are not easily reusable across deployments due to domain-specific tasks. Case studies reveal a convergent pattern across diverse domains (HR, infrastructure, analytics): establish golden question-answer sets, collect user interactions, and iteratively expand with expert review. This pattern extends to agent runtime through LLM-as-a-judge (§6.2), suggesting opportunities for reusable curation methods and synthetic data generation.

> **Finding 11:** Both studies show evaluation lacks standardization. 75% of cases evaluate without benchmarks relying on A/B testing and direct user feedback.

### 6.2. Evaluation Methods

Survey data shows human verification dominates evaluation despite automated alternatives. Seventy-four percent rely on human-in-the-loop, where domain experts, operators, or end-users validate the outputs for correctness (Figure 8).

Case study interviews reveal humans verify during both development and production. Development teams work with domain experts to create benchmarks and validate responses (e.g., `C15`: live systems incident configurations). In production, humans serve as final verifiers—agents suggest actions, experts approve execution (`C10`: engineers execute on SRE agent recommendation). One team forgoes all automated methods, hand-tuning agent configurations per client through feedback.

Model-based evaluation is second most common (52%), but survey data shows 39% combine it with human verification (Figure 10b). All case study teams using LLM-as-a-judge pair them with human review: judges score confidence for

each response, routing low-confidence cases to experts while randomly sampling some percentages of high-confidence outputs for ongoing validation (C01, C15).

Survey data shows human-in-the-loop (74%) and LLM-as-a-judge (52%) far exceed rule-based methods (42%). Case study interviews suggest this pattern reflects task complexity—customer support, HR operations, and compliance tasks require nuanced expertise that syntactic checks cannot capture. See §B.3 for detailed method descriptions and co-occurrence patterns.

> **Finding 12:** Human judgment dominates evaluation (74%), with LLM-as-a-judge (52%) as complement—revealing agents tackle tasks requiring nuanced expertise beyond rule-based verification.

# 7. *RQ4 Results:* What Are The Top Challenges In Deployment?

Survey data shows reliability remains the primary development bottleneck. Following the IEEE (1990) standard definition and prior works (Yao et al., 2025), we treat reliability as the probability of failure-free operation for a specified period in a specified environment. Thirty-eight percent of practitioners rank "Core Technical Performance"—encompassing reliability, robustness, and scalability—as their top priority in agent development, far exceeding governance (3%) or compliance (17%) (§B.4). We examine three key challenges: how teams achieve reliability, why evaluation remains difficult, and how teams manage security.

> **Finding 13:** Reliability remains the primary bottleneck.

## 7.1. Reliability Challenge

A paradox emerges: if reliability is the primary development bottleneck, how do agent systems reach production? We observe that teams mitigate reliability risks through strict environmental and autonomy constraints. A closer pass over our interviews surfaces three recurring reliability failure patterns. First, incomplete evaluation coverage forces teams to rely on expert review while they build task-specific test sets from scratch (C01, C03, C05, C09, C14, C15, C17). Second, correctness failures grow with task complexity, especially when systems combine heterogeneous or multimodal data, which often triggers human verification (C03, C06, C07, C10, C14). Third, legacy-system integration with existing security and compliance requirements limits functionality or narrows deployment scope (C03, C04, C10, C15, C16).

Interview data shows agents deploy in controlled environments minimizing failure impact: read-only mode where

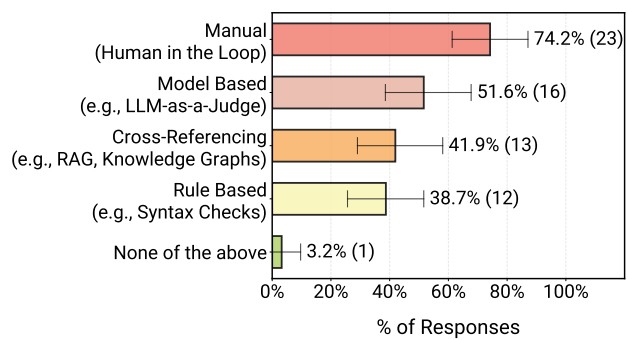

*Figure 8.* Evaluation methods reported by practitioners for deployed agentic systems ($N{=}31$, multi-select).

SRE agents generate bug reports for engineer review without modifying production (C16); sandbox verification where systems with write access undergo rule-based checks before production integration (C09, C11-C12); internal deployment serving employees where errors have lower consequences and expert oversight is immediate (§4.3); wrapper APIs restricting agents to abstraction layers that hide production system details (C16); and role-based access controls mirroring user permissions (C15).

Survey data reveals teams deliberately constrain autonomy. Sixty-eight percent execute fewer than 10 steps before human intervention (§5.4). Interviewed teams bound behavior through prompting and limited tooling. External-facing systems use particularly restricted workflows where trust and economic consequences demand control—pre-configured retrieval to specific document stores, mandatory human approval at critical steps, and fixed action sequences.

## 7.2. Evaluation Challenges

The patterns observed in §6.2, limited formal benchmarks and dominance of human evaluation (74%), stem from underlying technical challenges revealed in our interviews.

**Benchmark scarcity.** Interview data reveals three challenges with benchmark creation: (1) Regulated domains lack public data, forcing expensive expert-crafted datasets (C01, C16: months of data collection and labeling). (2) Client-specific customizations make standardized benchmarks infeasible (C04: proprietary toolsets and localized dialogue per deployment), leading teams to default to A/B testing and iterative client feedback.

**Real-world tasks are hard to verify.** Interview data reveals robust verification mechanisms don't always exist. Coding agents represent a rare case where verification occurs through compilation and test suites (C09, C12), enabling faster iteration. Most agents operate without fast automated signals. For example, insurance agents receive feedback only through delayed real consequences such as financial

losses or patient approval delays (C01). This verification gap may explain both the reliance on human evaluation and the concentration in productivity-focused applications (§4.1), where end-to-end time quantifies straightforwardly while harder-to-measure benefits remain underexplored.

### 7.3. Security and Privacy Challenges

Interview data reveals practitioners currently prioritize quality and correctness over security. Interviewed teams share that security is implicitly achieved through systems-level design (e.g., read-only agents) similar to §7.1 approaches. Current deployments also reflect this pattern, for example, 52% serve internal employees (§4.3). Systematic security mechanisms remain an open challenge as deployments expand to higher-stakes external settings.

Survey data shows agents frequently handle sensitive data, for example 69% retrieve confidential data (§B.4.1). Interview data reveals teams address privacy through contractual agreements with model providers preventing training on user data (C01: medical records).

## 8. Discussions & Conclusion

We provide the first systematic documentation of technical methods enabling real-world agent deployment. Through 20 in-depth case studies and surveys of 86 deployed systems, we make typically proprietary knowledge accessible to the research community. A central observation emerges from our data: **deployment techniques diverge from popular research methods**, signaling underexplored research opportunities. We consolidate our 13 findings (F) from RQ1-4 into 3 themes that reveal these research-practice gaps, which we hope these insights can support future agents research. We document production practice broadly and neutrally, reporting both confirmatory findings that establish a baseline for the field (e.g., productivity as the primary motivation, F1) and counterintuitive findings that challenge common research narratives (e.g., simple workflows, bounded autonomy, and frequent human oversight). We also go beyond high-level summaries: teams most often measure productivity through end-to-end completion time rather than model-centric metrics (§4.1).

**Surprising deployment patterns.** Model brittleness under provider updates drives practitioners away from post-training (F7) despite strong research results. This suggests opportunities in sample-efficient post-training and learning methods that transfer learned behaviors or preserve performance when base models update. Latency-tolerant applications (F4) indicate that inference-time scaling can satisfy agents' latency requirements, encouraging research in evolutionary search, test-verification architectures, and agent-specific batched inference scheduling. Multi-model

coordination (F6) creates opportunities for efficient multi-model inference, orchestration, and routing strategies.

**Reliability through system-level design**. Practitioners' preference for simple, controllable methods (F7, F9, F1) over blackbox optimization methods mirrors traditional software engineering practices. Our interviews reveal a distinct agent requirement for production reliability: continuous correctness verification during both development and runtime. This requirement creates research opportunities in observability (runtime monitoring, failure detection), evaluation (correctness metrics, real-world task benchmarks), and failure recovery. ML-focused research opportunities also include training models responsive to system specifications, learning under correctness specifications, and constrained optimization (verifiable actions, planning under bounds).

**Human-in-the-loop as design principle.** Practitioners deliberately incorporate human-in-the-loop as an architectural component, with agents executing limited steps before human intervention (F3, F9, F12). This currently addresses evaluation limitations (§7.2) and domain knowledge requirements. However, rather than viewing human-in-the-loop as a temporary constraint to overcome, our data suggests it has a potential to endure as a deliberate design principle. While much research emphasizes advancing single-agent autonomy, our findings reveal opportunities in agents augmenting human workers, suggesting research in agent workflows with human oversight, coordination interfaces, and methods for bounded autonomy within human-supervised systems.

As "agent engineering" emerges into a new discipline, MAP provides the research community with empirical insights about deployment realities to collectively advance agents that reliably deliver societal value on real-world tasks.

## Acknowledgements

We thank our many anonymous participants without whom beginning this study would not be possible. This research was supported by gifts from Accenture, Amazon, AMD, Anyscale, Broadcom Inc., Google, IBM, Intel, Intesa Sanpaolo, Lambda, Mibura Inc, Samsung SDS, and SAP. We also thank Alex Dimakis, Aditya Parameswaran, Drew Breunig, Tian Xia, Bhavya Chopra, and Shishir Patil for their valuable feedback and insightful discussions.

## Impact Statement

This paper presents work whose goal is to advance the field of Machine Learning. All participant data was collected under confidentiality agreements and reported in aggregated, de-identified form. There may be other potential societal consequences of our work, none which we feel must be specifically highlighted here.

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

## A. Organization of Appendix

The appendix is organized as follows. Appendix B provides supplementary results and deeper analyses that expand on the main findings, including extended breakdowns of agent requirements in *RQ1* (e.g., users, domains, and application constraints) in Appendix B.1. Appendix B.2 expands on *RQ2* results with the common architectural and engineering practices in production agents, including prompt construction strategies and recurring agentic workflow patterns. Appendices B.3 provides more detail on on evaluation practices that was discussed in *RQ3*. In addition, in Appendix B.4 we dig deeper into deployment challenges, with emphasis on security, privacy, and latency considerations.

Appendix C documents our study methodology, including the interview protocol and survey design, sampling, and data processing procedures. Appendix D reports results over the full survey dataset (beyond deployed agents only) and contrasts trends between deployed and non-deployed agents. Appendix E standardizes the terminology used throughout the paper (e.g., tasks, subtasks, and steps) and describes the workflow schema. Appendix F expands the related work discussion, situating MAP relative to prior surveys and empirical studies and clarifying how our study differs in scope and evidence. Finally, Appendix G contains the full survey instrument, including complete question text and branching logic.

## B. Supplementary Results and Analysis

In this appendix, we provide additional analysis and details related to the research questions discussed throughout the paper. We note that throughout the paper and appendix, for survey questions involving categorical comparisons, we report 95% confidence intervals computed using 1,000 bootstrap samples with replacement, where applicable.

### B.1. Agents application Requirements

#### B.1.1. AGENTS USERS

According to our study, the vast majority of agentic systems are designed to serve human users rather than other agents or software systems. Figure 3 shows the types and scale of users served by deployed agentic systems. As shown in this Figure, 92.5% of deployed agents report humans as their primary users. Among these, internal employees constitute the largest user group (52.2%), followed by external customers (40.3%). Only a small fraction of deployed systems (7.5%) primarily serve non-human consumers, such as downstream software or automated services.

Case study evidence suggests that the emphasis on internal users is often a deliberate deployment choice. Organizations frequently restrict early or initial deployments to internal settings to manage unresolved reliability, safety, and security risks. Internal deployments allow agent outputs to remain within organizational boundaries, where human oversight is readily available and errors typically carry lower external consequences. For example, several interviewed teams described internal operational agents that respond to employee requests, with human engineers able to intervene or override decisions when necessary. Across both internal- and external-facing systems, agents commonly support domain experts rather than operate autonomously. Many deployments require specialized domain knowledge to interpret agent outputs correctly—for instance, insurance authorization agents assisting nurses or incident response agents supporting site reliability engineers. This pattern reflects a broader role for agents as productivity-enhancing tools that augment expert workflows, with humans acting as final decision-makers or verifiers.

#### B.1.2. APPLICATION DOMAINS

Figure 2 summarizes the application domains of deployed Agentic AI systems in our survey data, spanning 26 distinct domains. These domains reflect a broad range of real-world use cases, from enterprise and software operations to healthcare, scientific discovery, and communication services. In this section, we describe how domain labels were derived from free-text survey responses and how infrequently occurring domains (outliers) were handled.

**Domain Normalization**    The survey response for the question related to agent application domains (QN4 in Appendix G.3) was collected in free-text form. Therefore, we normalized the domain responses to enable systematic analysis. We first applied the state-of-the-art semantic parser LOTUS (Patel et al., 2025) to perform semantic aggregation, deriving a set of candidate domain categories by grouping semantically similar phrases (e.g., healthcare, medical, patient monitoring) into unified labels (e.g., Healthcare Services). Based on the LOTUS output and as shown in Figure 2, this resulted in nine categories in total: eight high-level application domains and one residual "Other" category capturing long-tail responses.

*Table 1.* Survey Responses Recorded as 'Other' For Domain Analysis and Topic Normalization.

| | | | |
|---|---|---|---|
| chemical | proprietary-based networks | Telco | GTM Operations |
| supply chain | food & beverage industry | construction | automotive |
| travel | Advertising | Beauty & wellness | Privacy |
| entertainment & gaming | film & TV | social media | Paint industry |

Each survey response was then independently annotated by three annotators, who assigned one or more relevant domain labels from this LOTUS-derived category set. We measured inter-annotator agreement across all pairwise combinations of the three annotators and across all labels, yielding a mean Cohen's $\kappa$ of 0.636. This level of agreement is commonly interpreted as substantial for multi-class semantic labeling tasks, particularly given the multi-select nature of the question and the fact that many deployed agent systems naturally span adjacent or overlapping domains.

Disagreements primarily arose in borderline cases where systems plausibly fit multiple related categories (e.g., Corporate Services vs. Finance & Banking, or Technology vs. Research & Development), rather than from ambiguity in category definitions. In cases where no consensus was reached among the three annotators, a fourth expert annotator adjudicated the final label. Figure 2 reports results based on these final consensus annotations. Additional implementation and normalization details, including representative LOTUS programs, are provided below in this appendix section.

---

**Domain Normalization with LOTUS Semantic Aggregation**

```
df.sem_agg(
    f"Given user answers to a survey question in {QN4}, create a comprehensive bullet
        point list of answer categories. The survey question was: {header_map["QN4"]}."
)
```

---

**Outlier Domains**   Figure 2 includes an "Other" category, which captures the long-tailed distribution of categories not shown explicitly in the figure. Table 1 presents domains that were mentioned only once across all reported deployed agent use cases. These outlier domains highlight the long-tail diversity of agent applications beyond the dominant sectors observed in the main analysis, illustrating how agentic systems are being explored across a diverse set of real-world problems.

### B.1.3. LATENCY REQUIREMENTS

Relaxed latency requirements are commonly observed among deployed agents. Figure **??** shows the distribution of maximum allowable end-to-end latency. Minutes is the most common target, followed by seconds. Notably, 17.0% report no defined limit yet. The latency tolerance reflects the productivity focus from Section 4.1. Agents are often used to automate tasks that can take humans hours to complete. Consequently, an agent taking multiple minutes to respond remains orders of magnitude faster than non-agentic baselines. Interview participants emphasized this advantage: even if an agent takes five minutes, that remains more than $10\times$ faster than assigning the task to a person on the team, especially when staffing shortages exist and the task is secondary to the user's core responsibilities. Examples include nurses examining insurance details and software engineers responding to internal pager duty. Some deployed agents from case studies even batch requests hourly or overnight, further indicating latency is not a primary constraint.

However, this pattern breaks for real-time interactive applications. For example, practitioners building voice-driven systems report latency as their top challenge (Section B.4.2) during detailed case study. These systems compete against human conversation speeds rather than task completion baselines. Among our 20 detailed case studies, only 5 require real-time responsiveness. The remaining 15 cases tolerate extended processing times: 7 involve human review with relaxed timing, 5 operate as asynchronous background processes, and 3 have hybrid operation patterns. For these systems, processing times of minutes remain acceptable because the alternative can be days of human effort.

### B.2. Agents Architecture

### B.2.1. AGENTIC FRAMEWORKS

We find a divergence in framework adoption between survey respondents and interview case studies. As shown in Figure 9a, among deployed agents from the survey, two-thirds (60.7%) use third-party agentic frameworks. Reliance concentrates

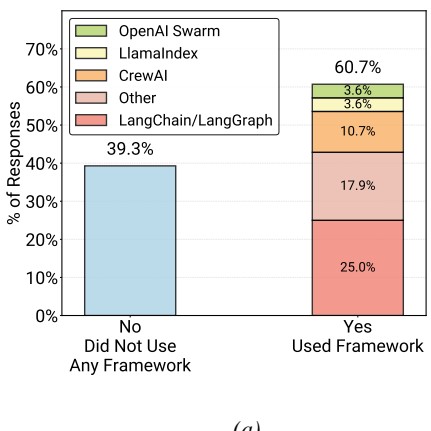 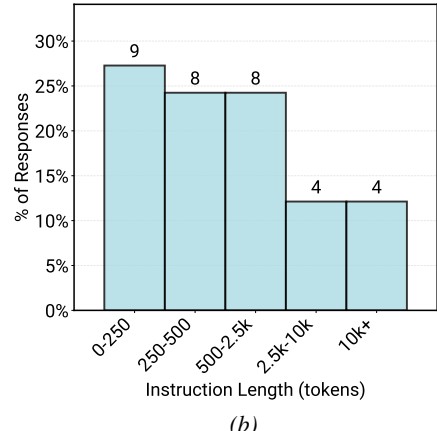

*(a)* *(b)*

*Figure 9.* Overview of core configuration and infrastructure choices in deployed Agentic AI systems. (a) Frameworks reported to use among teams building production agents ($N$=29). (b) Distribution of system prompt lengths measured in tokens ($N$=33).

around three primary frameworks: LangChain/LangGraph (lan; LangChain Inc., 2025) leads with 25.0%, followed by CrewAI (crewAI, 2025) at 10.7%, with LlamaIndex (LlamaIndex, 2025) and OpenAI Swarm (OpenAI, 2025b) both at 3.6%.

In sharp contrast, our detailed case studies reveal a strong preference for custom in-house agent implementations. Only 3 of 20 (15%) detailed case studies rely on external agent frameworks (2 use LangChain/LangGraph (lan; LangChain Inc., 2025), 1 uses DSPy (Khattab et al., 2024)). The remaining 17 teams (85%) build their agent application entirely in-house with direct model API calls. For example, one interview case explicitly shared that their agents are their own implementation of ReAct loops. Notably, two additional teams report starting with frameworks like CrewAI during the experimental prototyping phase but migrating to custom in-house solutions for production deployment to reduce dependency overhead.

We identify three core motivations for building custom solutions from the detailed case studies. First, *flexibility and control* are critical. Deployed agents often require vertical integration with proprietary infrastructure and customized data pipelines that rigid frameworks struggle to support. For example, one agent-native company deploys customer-facing agents across varied client environments, necessitating a bespoke orchestration layer. Second, *simplicity* drives the decision. Practitioners report that core agent loops are straightforward to implement using direct API calls. They prefer building minimal, purpose-built scaffolds rather than managing the dependency bloat and abstraction layers of large frameworks. Third, *security and privacy* policies sometime prohibit the use of certain external libraries in enterprise environments, compelling teams to develop compliant solutions internally.

### B.2.2. PROMPTING

As shown in Figure 6 Across deployed systems, prompt construction remains predominantly human-driven, with limited adoption of fully automated methods. We further examine system prompt length distributions among our survey deployed agents in Figure 9b. While a majority of agents use relatively concise prompts (51.5% under 500 tokens), prompt length exhibits a pronounced long tail. Among deployed agents, 24.2% use prompts between 500 and 2,500 tokens, 12.1% between 2,500 and 10,000 tokens, and an additional 12.1% exceed 10,000 tokens (Figure 9b). These longer prompts are rarely the result of a single design decision; instead, interview data suggests they often accumulate over time as systems incorporate guardrails, exception handling, policy constraints, and domain-specific instructions. As systems mature, prompts increasingly serve as centralized coordination artifacts rather than minimal task descriptions.

### B.3. Evaluation

This section provides additional detail on how practitioners evaluate deployed agentic AI systems. We complement the results for *RQ3* by exploring further on (i) whether deployed agents are explicitly compared against non-agentic alternatives, and (ii) the distribution of overlapping evaluation and verification strategies used in practice among deployed agents.

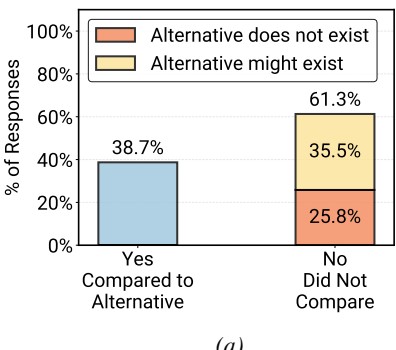 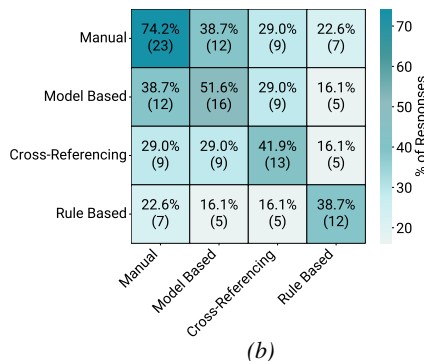

*(a)* *(b)*

*Figure 10.* Evaluation Practices in Agentic AI Systems: (a) Comparison to Alternatives: Shows whether participants explicitly compared their deployed agent against a non-agentic baseline (e.g., existing software, traditional workflows). (b) Evaluation Strategies Co-occurrence: Visualizes the pairwise overlap between evaluation strategies. Manual human-in-the-loop evaluation has the highest overlap with other strategies, suggesting that teams commonly rely on manual review to complement automated checks.

### B.3.1. COMPARISON TO BASELINES.

Figure 10a reports whether teams explicitly compare their deployed agentic systems against non-agentic baselines, such as existing software systems, traditional automated workflows, or human execution. 38.7% of respondents report conducting such comparisons, while the majority do not. In-depth interviews suggest several contributing factors: in some cases, agents are designed for tasks with no clear pre-existing alternative; in others, the baseline consists of a heterogeneous process involving multiple tools and human steps, making systematic technical comparison difficult. As a result, teams often evaluate agents based on task success, user outcomes, or qualitative improvements rather than direct performance deltas against a single baseline system.

### B.3.2. EVALUATION STRATEGIES OVERLAP.

As discussed in § 6.2 and Figure 8, human-in-the-loop evaluation (74.2%) and LLM-as-a-judge approaches (51.6%) are the most commonly adopted evaluation strategies among deployed agentic systems. We further dive deeper into analyzing different evaluation strategies in Figure 10b, where we explore which evaluation methods are commonly used together.

Figure 10b illustrates the co-occurrence of evaluation strategies across deployed agents, showing that human-in-the-loop evaluation is most frequently used in combination with other methods. Rather than relying on automated techniques in isolation, practitioners typically anchor rule-based verification, cross-referencing, and LLM-based judgments around human annotation. This pattern suggests that human judgment plays a central coordinating role in evaluation pipelines for production agentic systems.

This distribution indicates that many production agents operate in settings where correctness cannot be fully determined through deterministic rules or simple pattern matching. Instead, agents are deployed in domains requiring contextual understanding and nuanced judgment, such as customer support voice assistance or human resource operations. In these settings, practitioners rely on human and LLM-based evaluation to assess output quality, appropriateness, and task success, with automated checks serving a complementary role.

### B.4. Challenges

Like any emerging system, building agentic systems is not immune to challenges. We asked survey participants to rank the major categories of challenges they encounter during the development or operation of Agentic AI systems. Table 2 provides detailed descriptions of each challenge category identified in the survey, outlining the main technical and organizational issues practitioners reported when building Agentic AI systems. The five categories and their detailed definitions are provided in Table 2. Figure 11b illustrates how frequently each challenge category was assigned a given rank. For example, 'Core Technical Performance' was ranked as the most significant challenge (#1) by 37.9% of respondents, by far more than any other category, indicating it remains the dominant source of difficulty in current Agentic AI system development. 'Core Technical Performance' encompasses a wide range of issues, including robustness, reliability, scalability, latency, and resource constraints. Its prevalence suggests that much of the community's current effort is devoted to ensuring that systems perform consistently and dependably under real-world conditions. Following closely as shown in Figure 11a 'Data and

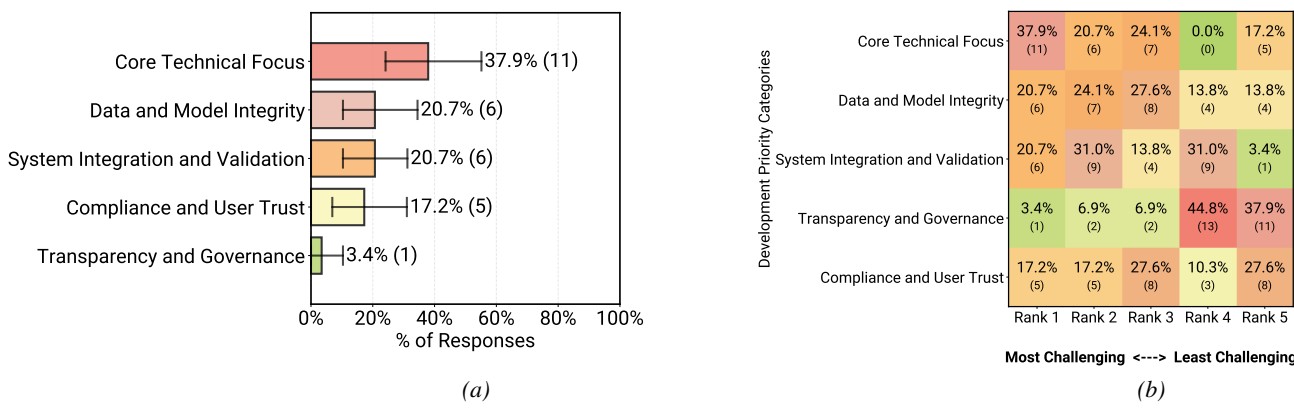

*Figure 11.* Major challenges encountered across agents deployment ($N = 29$). (a) Distribution of top-ranked (Rank 1) challenges reported for deployed agentic systems. Lower-ranked categories reflect areas that respondents perceived as *lower priority relative to other challenges*, rather than challenges that are fully resolved or unimportant (e.g., compliance and governance). (b) Heatmap showing how frequently each challenge category was assigned to different difficulty ranks (1 = most challenging, 5 = least challenging) across deployed agents. Overall, the results indicate that *Core Technical Focus* remains the dominant source of friction in current deployments.

*Table 2.* Major categories of challenges reported by participants.

| Challenge Category | Representative Issues and Focus Areas |
|---|---|
| **Core Technical Performance** | Robustness and reliability—ensuring consistent, correct behavior in diverse and unpredictable environments; scalability—supporting growth in users, data, and tasks without performance degradation; real-time responsiveness—meeting latency and timing requirements; resource constraints—managing compute, memory, and energy efficiently. |
| **Data and Model Integrity** | Data quality and availability—access to clean, timely, and relevant data; model and concept drift—adapting to changes in data distributions and task definitions; versioning and reproducibility—tracking models, data, and configurations for auditability. |
| **System Integration and Validation** | Integration with legacy systems—connecting with existing infrastructure and APIs; testing and validation—simulating and verifying agent behavior before deployment; security and adversarial robustness—defending against manipulation and exploitation. |
| **Transparency and Governance** | Explainability and interpretability—making decisions understandable to humans; bias and fairness—preventing discriminatory or unjust outcomes; accountability and responsibility—clarifying who is liable for agentic decisions. |
| **Compliance and User Trust** | Privacy and data protection—ensuring adherence to data regulations (e.g., GDPR); user trust and adoption—building confidence through transparency and reliability; regulatory compliance—meeting legal standards for autonomy, safety, and transparency. |

Model Integrity' and 'System Integration and Validation' were reported as second-ranked persistent sources of friction when transitioning systems from research prototypes to production environments. In contrast, 'Transparency and Governance' and Compliance and User Trust were ranked as lower-priority concerns, indicating that while practitioners recognize its long-term importance, it is not yet perceived as a primary bottleneck in current development cycles.

### B.4.1. Security and Privacy Challenges

Security and privacy consistently rank as secondary concerns in both of our studies, with practitioners prioritizing output quality and correctness. Figure 11a shows that Compliance and User Trust ranks fourth among challenge categories. Given that §4.3 shows 52.2% of systems serve internal employees and many systems with human supervision, this prioritization reflects current deployment environments and requirements rather than dismissing security's importance.

**Data ingestion and handling.** Survey results in Figure 12a show that 89.7% of systems ingest information from databases, 65.5% ingest real-time user input, and 51.7% ingest other real-time signals. Notably, 69.0% of systems retrieve confidential or sensitive data, while only 34.5% retrieve persistent public data. Given the high prevalence of sensitive data usage and user inputs, preserving privacy is critical. Our interview case studies reveal that teams address this through legal methods.

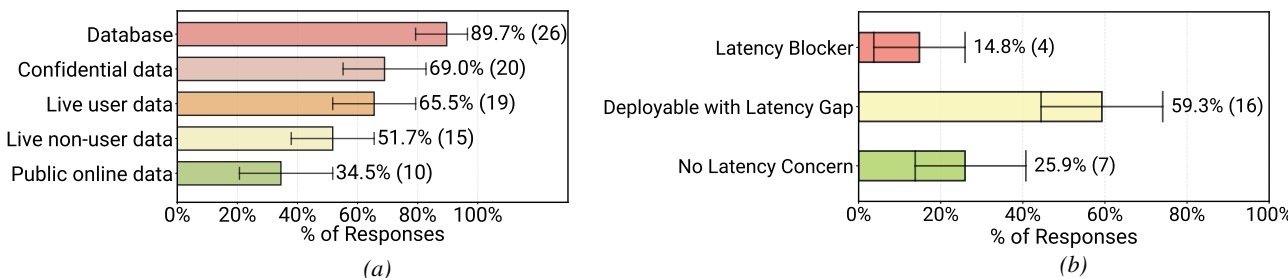

*Figure 12.* Supplementary deployment characteristics of agentic systems ($N = 27$–$29$). (a) Overview of data ingestion and handling capabilities in deployed agents. The question was multi-select, allowing participants to indicate all data handling methods integrated into their systems. The distribution highlights a strong reliance on internal infrastructure over public data sources. (b)Degree to which latency is reported as a deployment challenge. The results suggest that latency is rarely a strict blocker for most deployed agentic systems.

For example, a team building healthcare agents report relying on standard data-handling practices and strict contractual agreements with model providers to prevent training on their user data.

**Security practices.** In-depth interview participants describe four approaches to managing security risks through *constrained agent design*. First, six teams restrict agents to "read-only" operations to prevent state modification. For example, one SRE agent case study generate bug reports and proposes action plans, but leaves the final execution to human engineers. Second, three teams deploy agents in sandboxed or simulated environments to isolate live systems. In one instance, a code migration agent generates and tests changes in a mirrored sandbox, merging code only after software verification. Third, one team builds an abstraction layer between agents and production environments. This team constructs wrapper APIs around production tools, restricting the agent to this intermediate layer and hiding internal function details. Finally, one team enforces role-based access controls that mirror agent user's permissions. However, the agent team reports this remains challenging, as agents can bypass these configurations when accessing tools or documents with conflicting permissions.

### B.4.2. LATENCY CHALLENGES

We examine the degree to which agent execution latency hinders deployment. Survey results indicate that latency represents a manageable friction rather than a hard stop for most teams. Figure 12b shows that only 14.8% of deployed survey agents identify latency as a critical deployment blocker requiring immediate resolution, while the majority (59.3%) report it as a marginal issue, where current latency is suboptimal but sufficient for deployment. We suspect that this tolerance correlates with the prevalence (15/20 in detailed interview case studies) of asynchronous agent execution paradigm (Section B.1.3) and (52.2% from survey) internal user bases (Section 4.3). Notably, we observe a consistent latency distribution across the full survey dataset, including experimental systems (Figure 24b). We believe this consistency signals a broader preference for building offline agents, as discussed in Section B.1.3.

**Interactive agent latency requirements.** While latency is not a critical challenge for most agent applications, it remains a critical bottleneck for real-time interactive agents. Two interviewed teams, building voice agents, report continuous engineering efforts to match human conversational speeds. Unlike asynchronous workflows, these systems require seamless turn-taking where delays disrupt the user experience. Achieving fluid real-time responsiveness beyond rigid turn-based exchanges remains an open research question and development challenge.

**Practical latency management.** Interview participants describe two approaches to managing latency. First, teams commonly implement hard limits on maximum steps or model inference calls, typically derived from heuristics. Second, one team adopts a creative solution by pre-building a database of request types and agent actions (tool calls), then employing semantic similarity search at runtime to identify similar requests and serve prebuilt actions, reducing response times by orders of magnitude compared to reasoning and generating new responses. These workarounds demonstrate that practitioners currently rely on system-level engineering to bypass the inherent latency costs of foundation models.

### B.4.3. MODALITIES

Supporting multiple data modalities emerges as an additional deployment challenge for production agent systems. Beyond text-based interactions, practitioners increasingly aim to extend agents to handle richer inputs and outputs, including speech, images, video, spatiotemporal data, and domain-specific scientific formats. While these capabilities unlock new application

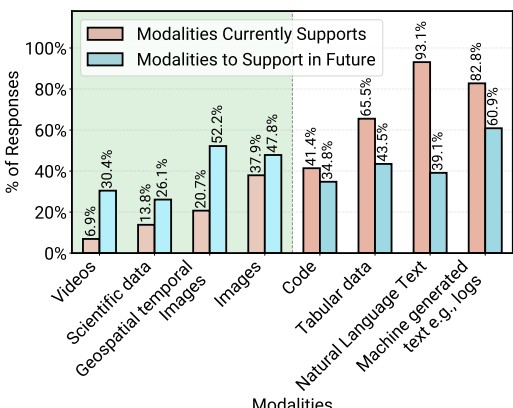

*Figure 13.* Data modalities already supported (red) versus modalities planned for future support (blue) in production agent systems ($N$=29). Bars to the left of the dashed line indicate modalities with expected increases in future support, whereas modalities to the right are already widely supported with limited planned expansion. Interestingly, the modalities with the largest planned growth are all non-textual, pointing toward increasingly multimodal agent systems.

opportunities, they also introduce substantial engineering, evaluation, and reliability challenges.

To understand current practice and future direction, we asked survey participants to report (1) which data modalities their deployed agents currently support and (2) which additional modalities they plan to add. Figure 13 summarizes these responses for deployed agent systems, contrasting modalities already supported (red) with those planned for future support (blue). As shown in this Figure, text dominate current deployments: 93% of surveyed production agents accept natural-language text input. In contrast, support for non-textual modalities remains relatively limited today. However, the strongest expected growth is concentrated precisely in these less mature modalities. Bars to the left of the dashed line indicate modalities with anticipated increases in future support, while modalities to the right are already relatively more adopted with comparatively limited planned expansion. Notably, the modalities with the largest projected growth—such as images, video, spatiotemporal data, and scientific data—are all non-textual, pointing toward increasingly multimodal agent systems.

Interview data helps contextualize this pattern. Participants consistently report that early deployments prioritize modalities that are easy to measure, validate, and debug. Text-based agents benefit from established evaluation workflows, human verification, and relatively fast iteration cycles. In contrast, multimodal agents face weaker correctness signals, higher infrastructure complexity, and more expensive data pipelines. As a result, teams often defer multimodal expansion until text-centric deployments achieve sufficient reliability.

This trend presents both challenges and opportunities. On the challenge side, multimodal agents complicate evaluation, observability, and failure detection, exacerbating issues already present in text-only systems. On the opportunity side, growing practitioner demand suggests a clear need for research on multimodal agent architectures, modality-aware evaluation methods, and system designs that support reliable multimodal execution at runtime. As production agents mature, we expect modality expansion to follow successful stabilization of text-based deployments, extending agent capabilities toward richer, more complex real-world inputs.

## C. Collected Data Details

In this section, we provide additional details about our collected data sources, including (i) the survey respondent population and the deployment characteristics of the reported agentic systems (Appendix C.1), and (ii) supplementary information on the in-depth interview case studies such as demographics and organizational context (Appendix C.2). In total, our survey collected 306 valid responses and, together with 20 in-depth interviews, this data forms the empirical basis of this study on AI agents in production.

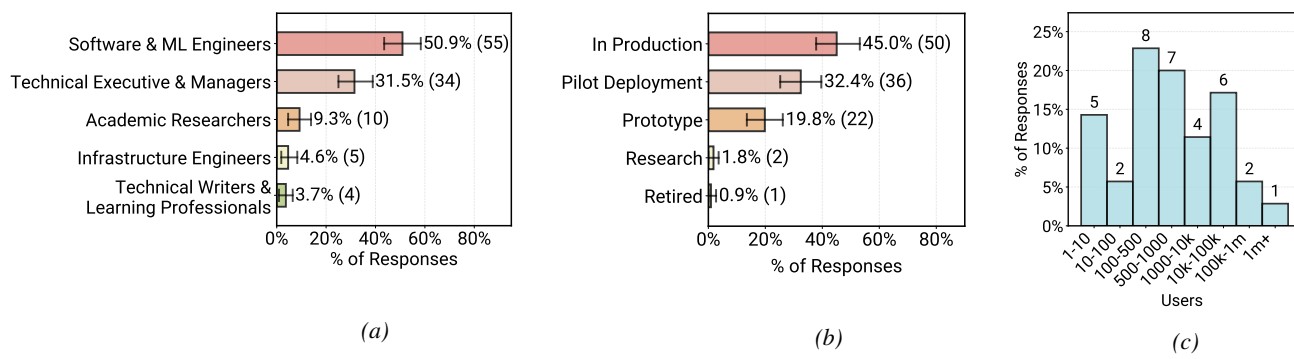

*Figure 14.* Overview of survey respondent and system characteristics across all agents the survey data: (a) roles of survey participants by primary contribution area ($N{=}108$), (b) deployment stages of Agentic AI systems ($N{=}111$) that survey participants contributed to, and (c) reported number of end users ($N{=}35$) for the Agentic systems survey participants contributed to.

*Table 3.* Case studies grouped by application domain. There are 20 cases total, but similar cases are merged for clarity and confidentiality.

| Business Operations |
|---|
| C01: Insurance claims workflow automation |
| C02: Customer care internal operations assistance |
| C03: Human resources workflow automation and assistance |
| **Communication Tech, Multi-lingual Multi-dialect** |
| C04: Communication automation services |
| C05: Automotive communication services |
| **Scientific Discovery** |
| C06: Biomedical sciences workflow automation |
| C07: Materials safety and regulatory analysis automation |
| C08: Chemical data interactive exploration |
| **Software DevOps** |
| C09: Spark version code and runtime migration |
| C10: Software development life cycle assistance end-to-end |
| C11: Software engineer/developer slack support |
| C12: SQL optimization |
| C13: Code auto-completion and syntax error correction |
| **Software & Business Operations** |
| C14: Data analysis and visualization |
| C15: Enterprise cloud engineer and business assistance |
| C16: Site reliability incident diagnoses and resolution |
| C17: Software products technical question answering |

## C.1. Survey Data Details

Survey respondents self-identified as practitioners actively building AI agents across 26 application domains (Figure 2), spanning areas such as finance, healthcare, and legal services. Of the 306 respondents, 294 reported that they have directly contributed to building and designing at least one agent system. As reported in Figure 14a ($N{=}108$), among those who disclosed their role, respondents are predominantly technical professionals, with a large share identifying as software and machine learning engineers. Figure 14b reports the deployment stages of the agentic systems respondents contributed to ($N{=}111$); among those who reported deployment stage, 82% indicated that their systems are in *production* or *pilot* phases, reflecting rapid transition from experimental prototypes to real-world deployments.

Beyond deployment stage, we examine the scale of the user base for deployed systems. Figure 14c summarizes reported end-user counts ($N{=}35$), showing substantial variation in deployment scale. In particular, 42.9% of reported deployments serve user bases in the hundreds, while 25.7% serve tens of thousands to over one million daily users. Together, these distributions indicate that our survey captures both smaller deployments and high-impact systems operating at significant scale, motivating our focus on deployed agents in the main analysis.

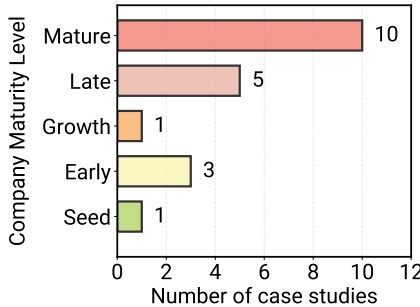

*Figure 15.* The distribution of source institution maturities across in-depth interview-based case studies. The minority (5/20) are from seed-stage startups (validating product-market fit), early-stage startups (proving scalable business models), and growth-stage startups (rapidly expanding market share and operations). The majority (15/20) are from late-stage and mature institutions (with established market positions). The stages are approximated from limited public information e.g. size, sector, and annual recurring revenue.

## C.2. In-Depth Case Study Details

This appendix provides details on the in-depth case studies to support the qualitative findings. In total, we curated 20 interview-based case studies, selected to reflect diversity in application settings, organizational maturity, and geographic reach. The anonymized case studies and their representative use-case descriptions are summarized in Table 3. These cases span multiple application categories, including business operations (`C01-C03`), communication technologies and multilingual or multidialect systems (`C04-C05`), scientific discovery (`C06-C08`), software DevOps and infrastructure (`C09-C13`), and software and business operations (`C14-C16`). Each case is referenced throughout the paper using anonymized identifiers (`C01`, `C02`, ...). When organizations operated multiple agent deployments, we prioritized selecting distinct use cases to avoid over-representing any single institution and to capture a broader range of deployment patterns.

### C.2.1. DEPLOYMENT STAGE AND ORGANIZATIONAL MATURITY

All interviewed systems serve real-world users: 14 cases are in full production and 6 are in final pilot phases. The studied systems support both internal users (5 cases) and external enterprise users (15 cases), and originate from organizations spanning a wide range of maturity levels—from seed-stage startups to large enterprises with global footprints (Figure 15). To respect confidentiality agreements with case study sources, we report only aggregate statistics about organizational characteristics and geographic footprint.

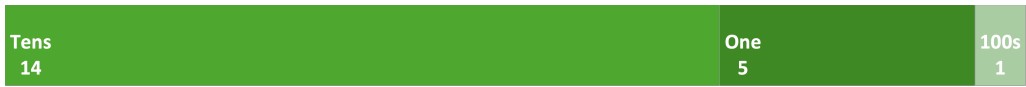

*Figure 16.* Case study sources are present in one to hundreds of countries. This shows the distribution of cases by sources' country spread.

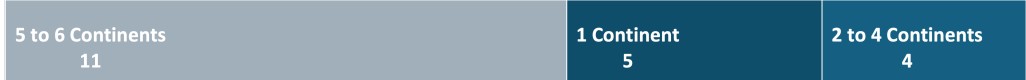

*Figure 17.* Case study sources are present in 1 to 6 continents. This shows the distribution of cases by sources' continental spread.

### C.2.2. GEOGRAPHIC DISTRIBUTION.

Figures 16 and 17 summarize the geographic footprint of organizations in our case studies. As shown in Figure 16, case study sources operate across a wide range of country-level presence, from organizations active in a single country to globally distributed deployments spanning hundreds of countries. Figure 17 shows that case study sources operate across one to six continents, indicating that the interviewed systems include both regionally focused deployments and globally distributed services. These distributions suggest that the qualitative findings reflect agent deployments operating under heterogeneous regulatory, linguistic, and operational environments, rather than being confined to a single geographic context.

C.2.3. PARTICIPANT BACKGROUND, RECRUITMENT, AND DATA COLLECTION TIMELINE

We conducted interviews with technical practitioners directly responsible for the design, implementation, or operation of the studied agentic systems, including ML engineers, software engineers, and senior technical leads. To protect confidentiality, we do not report individual-level demographic attributes; however, participants represented a range of engineering roles and experience levels spanning system architecture, deployment, evaluation, and ongoing operations. Although studied systems may have been publicly announced, public-facing materials (e.g., product documentation, marketing releases, or high-level technical overviews) do not capture the implementation- and operations-level detail central to our analysis. As such, access to practitioners with the required depth of expertise typically occurs through professional networks and practitioner-oriented events. Participants were therefore recruited via the authors' professional networks as well as outreach through presentations at agent-focused technical venues (anonymized), and were screened to ensure active, hands-on involvement in the development or maintenance of the systems under study.

**Data collection occurred across multiple phases**: initial survey and interview design began in March, followed by interviews with upstream stakeholders from mid-April through mid-May. Three subsequent recruitment rounds were conducted through agent-focused technical events spaced roughly two months apart—late May, early August, and mid-October—with the final interview completed in November.

**Considerations for fast-evolving systems.** Because agentic systems and their supporting infrastructure evolve rapidly, collecting data across multiple phases allowed us to capture changes in engineering practices as they emerged over the study period. This staggered timeline also broadened the range of contexts represented, as participants engaged with different system states, model versions, or operational conditions. However, we recognize that later interviews may reflect different technical environments than earlier ones, which can complicate synthesis and introduce recency-related bias. To mitigate these issues, we analyzed interviews with close attention to their temporal context, distinguishing themes that appeared consistently across phases from those tied to time-specific system developments.

## C.3. Interviews Details

C.3.1. STANDARDIZED INTERVIEW PROCEDURES.

We followed a consistent sequence of pre-, in-, and post-interview procedures across all case studies. Interviews adhered to a shared semi-structured protocol spanning 11 topic areas (Appendix C.3.2), covering system architecture, evaluation practices, deployment challenges, operational constraints, and measures of agent value. Each interview lasted 30–90 minutes and involved 2–5 participants with fixed roles to improve consistency and reduce interviewer effects. Interviewers were selected to maintain organizational neutrality with respect to participating teams.

Pre-interview context gathering, when applicable, was restricted to publicly available sources (e.g., product documentation or engineering blogs) to avoid introducing private or leading assumptions. Depending on participant preference, interviews were either recorded or documented through detailed human notes. Post-interview summaries were cross-validated among interviewers to ensure accuracy and internal consistency. In accordance with confidentiality agreements, all data are anonymized and reported only in aggregate. Interview discussions were guided by the predefined topic groups, with interviewers instructed to prioritize topics 1–5, followed by 6–8, and then 9–11 as time permitted. Topics with available public information were used primarily for verification rather than open-ended elicitation.

C.3.2. INTERVIEW OUTLINE

1. **The root problem (benefit) the system is addressing (providing):** What is the ultimate benefit? What is the system replacing and why?

2. **Key success metrics and evaluation mechanism:** What tools, techniques, systems, etc. are used to ensure the system meets user and stakeholder objectives? Is data corresponding to the expected or past system behavior available for the evaluation?

3. **Key aspects of the system design and implementation:** What programming framework was used? What is the general architecture? What are the steps, stages, and cycles? How are common components (e.g. routers, LLM-as-a-Judge, other verifiers, HIL) combined and why? What is the ratio of automation to human interaction and why—by design or limitation?

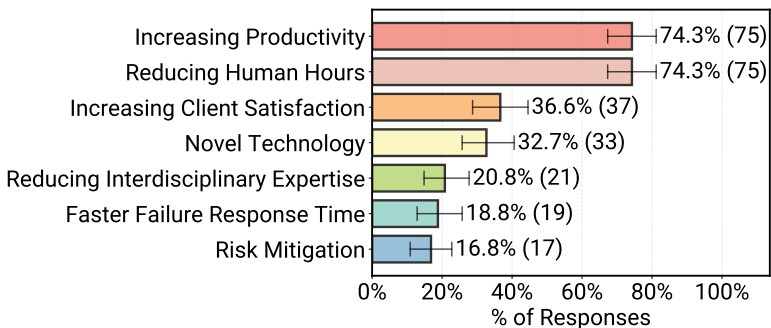

*Figure 18.* [All Data] Reasons practitioners build AI agents across all development stages (Production, Pilot, Prototype, and Research). Increasing productivity remains the most selected benefit across the full dataset ($N = 101$).

4. **The state of the system or its development:** Is the system in production, or was it never meant for production (purely for AI research, learning, upskilling)? Was the system prototyped for production but abandoned—why, and what were the critical limitations? Were there surprises in the development or evaluation process? Did some things work better or worse than expected, and if so, what?

5. **Known constraints or requirements of end-users and stakeholders:** What are the security, confidentiality, regulatory, latency, SLO/SLA, or other requirements?

6. **Advantage of an agentic AI system solution over alternative approaches:** Do reasonable alternative solutions exist for this problem, or is this a novel solution made possible with Agentic AI? Against existing alternatives, has comparative analysis been conducted? What are the comparative benefits, costs, and return on investment (ROI)?

7. **System dependencies and complexity:** what is the quantity, quality, and availability of tools and data for verification and generation?

8. **End-user quantity, expertise levels, and organizational domains.** is it a product for internal-use only or public external use? Does it support multiple institutions? Are there institution-specific or regulatory boundaries limiting the quantity of users? Are target users domain experts or novices? How many of each user group are there and how many are targeted (order of magnitude)?...

9. **Estimated cost versus value or benefit:** What is the estimated cost (sunk and expected ongoing costs) of developing and operating the system versus the estimated value or benefit? Is the respondent aware? What is the value, how is ROI being calculated?

10. **System stakeholders:** Who ultimately benefits from deployment? Who is impacted by safety, security, etc. failures and limitations? What is the expected impact on the company/institution (e.g. reduced hiring, retraining, broader user-base etc.)?

11. **Your role and activities:** What is your role in the development of the agentic AI system(s) you are describing?

## D. Results on All Survey Data

In the main body of the paper, we focused on results filtered exclusively to *deployed agents* in production or pilot phases, to highlight successful real-world practices under realistic operational constraints. In this appendix, we present the corresponding results computed over [All Data]: all 306 valid survey responses, regardless of deployment stage. This expanded view includes prototype, research, and legacy systems (sunset and retired), providing a broader perspective across the full agent development lifecycle. For ease of comparison, each [All Data] figure mirrors a figure in the main text and uses the same layout and question wording. In the discussion below, we briefly describe the key patterns in the full dataset and highlight how they compare to the deployed-only subset.

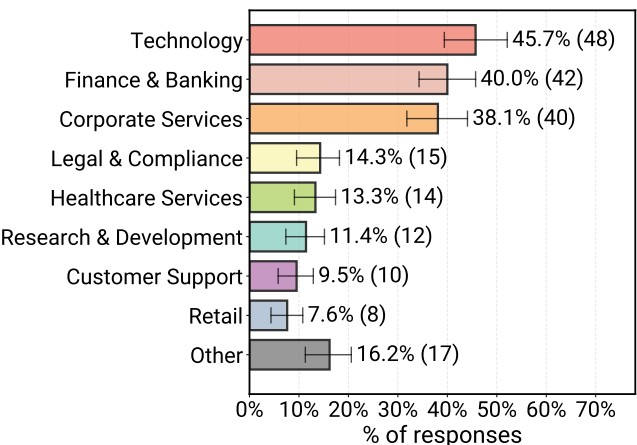

*Figure 19.* [All Data] Application domains where practitioners build Agents across all development stages ($N = 105$). This is a multi-class question where each system may be assigned to multiple domain categories; proportions therefore do not sum to 1.

### D.1. *RQ1* Agents Applications

**Why Agents?**   Figure 18 reproduces our analysis of motivations for using agents over non-agentic alternatives using all survey responses. We observe that the overall ranking of benefits is highly stable compared to deployed only agents as shown in Figure 1: increasing productivity remains the most frequently selected reason for adopting agents, followed by reducing human hours .

**Application Domains.**   For application domains of agents, (Figure 19) become even more diverse in the full dataset compared to deployed only agents (Figure 2). The same high-level industries i.e., finance and banking, technology, and corporate services remain prominent. However, the long tail of "other" domains grows, reflecting additional experimental and research systems in areas such as education, creative tools, and scientific workflows that have not yet reached deployment.

**End users**   Consistent with the deployed-only subset, the vast majority of systems in the full dataset still target human users as shown in Figure 20a, with internal employees and external customers together comprising most end-user bases. The relative proportions shift only slightly when we include prototypes, suggesting that human-centric interaction remains the default even in early experimentation.

**latency requirements.**   Latency requirements also similarly remain relaxed in the full dataset (Figure 20b compared to Figure ??). Most teams still report tolerating response times on the order of minutes, with a non-trivial fraction indicating that no explicit latency limit has been set. Compared to deployed agents only, the fraction of agents with undefined latency budgets is slightly higher in [All Data], which is consistent with prototypes and research artifacts that have not yet been hardened with production SLOs. Overall, these figures confirm that the preference for latency-relaxed, quality-focused applications is not an artifact of our deployment filter.

### D.2. Models, Architectures, and Prompting

**Autonomy and Architechture**   Figure 21 explores *RQ2*, focusing on core component configurations and agent architectures across [All Data]. Compared to deployed agents (Figure 21b), the distribution of the number of distinct models used exhibits a heavier tail: non-deployed and research systems are more likely to combine many distinct models, leading to a higher incidence of configurations with four or more models. This pattern aligns with our qualitative observation that teams explore richer multi-model setups during early experimentation and then consolidate to a smaller, more manageable set of models as they move toward deployment.

A comparison between Figure 7a and Figure 21a reveals a clearer separation in autonomy. When we include all agents, systems allow a greater number of autonomous steps or cycles before human intervention compared to deployed agents only. Experimental and research systems are more likely to fall into the "tens of steps" and "no explicit limit" regimes, whereas production deployments concentrate in the low-step regime to control cost, latency, and failure amplification. Taken together,

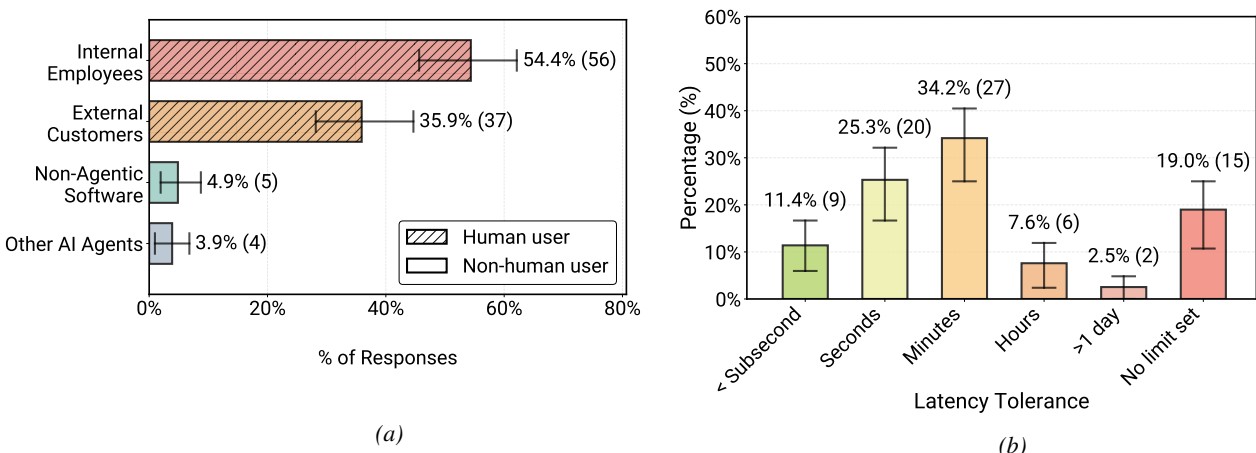

*(a)*

*(b)*

*Figure 20.* [All Data] Overview of Agentic AI system characteristics across all development stages in terms of primary end users and latency requirements. (a) Distribution of primary end users ($N$=103), where hatched bars (////) denote human end-users (corresponds to Figure 3); and (b) Reported tolerable end-to-end response latency for all systems ($N$=84) which corresponds to Figure **??** on deployed agents. The high percentage of human end-users and tolerance for minute-level latency are consistent across the full dataset.

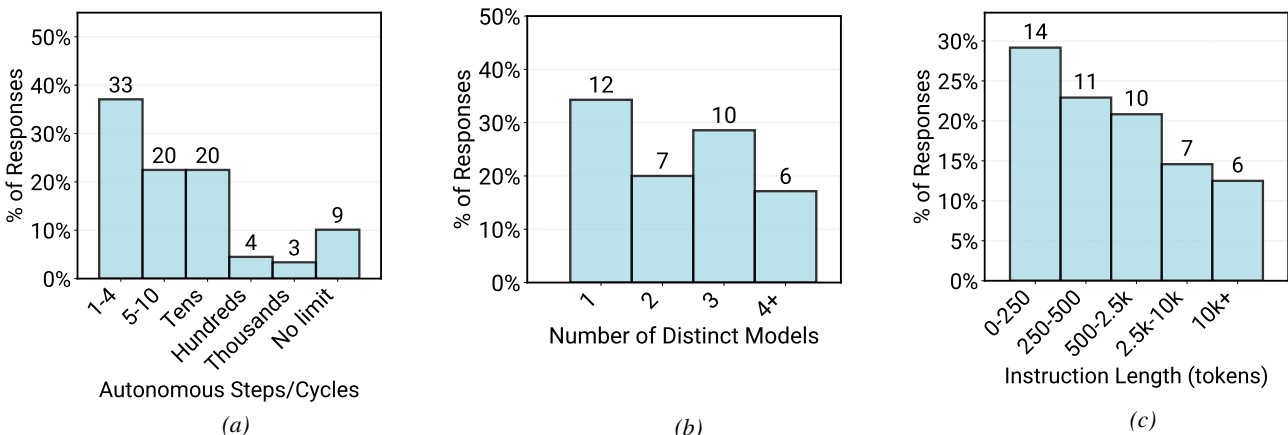

*(a)*

*(b)*

*(c)*

*Figure 21.* [All Data] Overview of core component configurations and architectures across all agents. (a) Number of autonomous execution steps before user intervention ($N$=89). Results on all survey data allow for a greater number of steps before human interference compared to deployed agents only (Figure 7a), reflecting stricter control and monitoring needs in production environments. (b) Number of distinct models combined to solve a single logical task ($N$=35). Deployed agents (Figure 7b) tend to use fewer models than the full dataset, which includes more experimental systems using a higher number of distinct models; and (c) Distribution of prompt lengths in tokens ($N$=48). Prompt length remains similar across deployed-only (Figure 9b) and all-data subsets.

these results reinforce our interpretation that bounded autonomy is a deliberate design choice for production reliability, while higher autonomy is more common in exploratory settings.

**Frameworks.** Figure 22b compares framework usage across all agents and corresponds to Figure 9a in the main text. The overall split between using a framework versus no framework remains nearly identical between the full dataset and the deployed-only subset, indicating that teams decide early whether to invest in a framework-based stack or implement their own orchestration. Within the "framework" group, the `Other` category expands slightly in [All Data], reflecting experimentation with a broader variety of less common or homegrown frameworks during research and prototyping stages, beyond the dominant framework families.

**Prompting.** Figure 22a reproduces our analysis of prompt construction strategies using all survey responses. The results show that human-crafted prompts remain central across the full dataset: fully manual and manual+LLM strategies continue to be the dominant modes. However, when we include non-deployed agents, we observe a modest shift toward more automated prompting. Fully manual prompting is slightly more common among deployed agents (Figure 6), while the [All

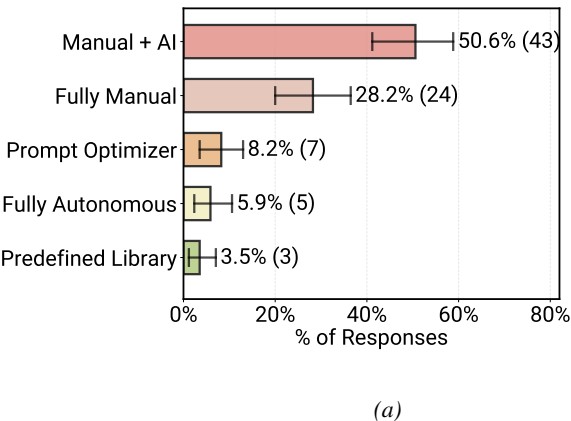
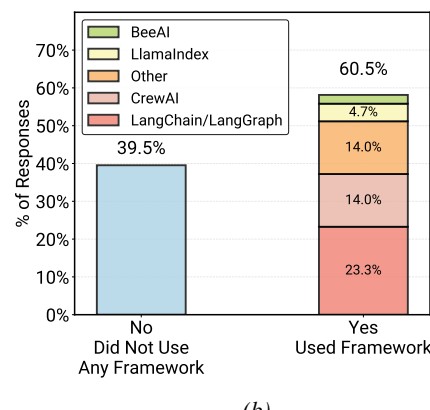

*(a)*                                                                *(b)*

*Figure 22.* [All Data] Overview of core technical implementations: Prompt Construction and Framework Usage. The left and right figures correspond to Figure 6 and Figure 9a in the main text respectively but they include all survey data. (a) Distribution of prompt construction strategies across all agents ($N$=85). Human input remains central to prompt crafting, however, fully manual is slightly more common in deployed agents (Figure 6) compared to all data in the left plot. Similarly 'fully autonomous' prompting is more common in the All Data, suggesting that manual prompting is more favored for production systems. (b) Frameworks reported to support critical functionality ($N$=43). The percentage of practitioners using a framework versus not using one remains almost exactly the same between the full dataset (right plot) and the deployed-agents-only (Figure 9a). The "other frameworks" category increases slightly in the full dataset compared to deployed agents only, likely reflecting more diverse experimentation in non-production systems.

Data] distribution shows somewhat higher use of "fully autonomous" prompting and prompt optimizers. This suggests that automated prompt construction is currently used more as an experimental technique and is less frequently adopted in production systems, where controllability is critical.

Prompt lengths remain broadly similar between the deployed-only dataset (Figure 9b) and the full dataset (Figure 21c).

### D.3. Evaluation Practices for All Agents

This section provides more details on evaluation practices, focusing on *RQ3* across [All Data] . Figure 23 presents evaluation practices across all agents and mirrors Figure 8 and Figure 10 in the main text. Figure 23a shows distribution of comparison against non-agentic baselines. When we include prototypes and research systems, the fraction of teams that explicitly compare their agents to alternative solutions is slightly *lower* than in the deployed-only subset (34% in all data vs. 38.7% in deployed agents in Figure 10a). This suggests that some experimental agents are perhaps still in early stages where rigorous baseline comparison has not yet been prioritized, or where teams are primarily exploring feasibility rather than relative gains.

Figure 23b reports the distribution of different evaluation methods. The ordering of methods remains unchanged: human-in-the-loop (manual) evaluation is still the most common strategy, followed by model-based evaluation (LLM-as-a-judge). However, manual evaluation is somewhat more prevalent among deployed agents (Figure 8), whereas the [All Data] distribution shows a relatively higher share of automated methods. This is consistent with the idea that experimental and research systems may rely more on automated or lightweight checks, while production systems invest more heavily in human verification before and during deployment.

Figure 23c visualizes co-occurrence patterns between evaluation strategies. Human-in-the-loop evaluation remains the central hub in the evaluation graph, with high overlap with all other methods in the full dataset. At the same time, its co-occurrence with other strategies is slightly lower in [All Data] than in the deployed-only subset (Figure 10b), reflecting that some experimental systems use model-based or rule-based checks without consistently pairing them with human review. In contrast, deployed agents are more likely to combine automated evaluation with human verification perobably for higher assurance.

### D.4. Challenges Across All Agent Deployment Stages

In this section, we focus on challenges encountered when building agent systems across different deployment stages, comparing deployed agents with non-deployed (i.e., prototype or research) agents in [All Data]. Specifically, we examine challenges related to data handling, latency, and modality support across all agents.

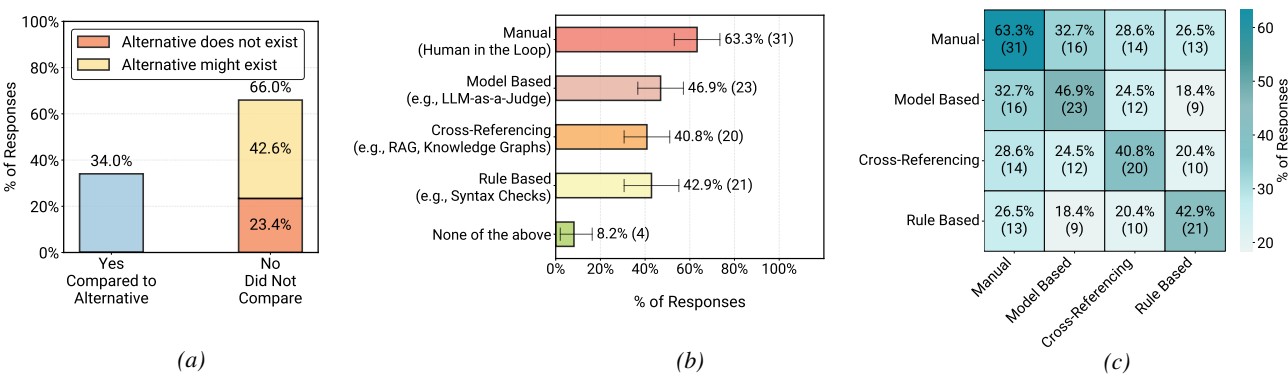

*(a)*        *(b)*        *(c)*

*Figure 23.* [All Data] Evaluation Practices in Agents. This figure corresponds to Figure 8 and 10 but includes all survey data ($N = 47$). (a) Comparison to Alternatives: Shows whether participants explicitly compared their agent against a non-agentic baseline. Deployed agents (Figure 10a show a higher comparison rate (38.7% in deployed vs. 34% in all data) of comparison to alternative solutions, suggesting that experimental prototypes may not have invest as much on evaluation stages yet. (b) Evaluation Methods Distribution: Distribution of different evaluation strategies reported by survey participants. *Manual evaluation (human-in-the-loop)* is used more for deployed agents (Figure 8) compared to the full dataset, which includes more experimental and research systems. Autonomous evaluation methods are relatively less common in deployed agents compared to the data for all agents in oru survey. (c) Evaluation Strategies Co-occurrence: Visualizes the pairwise overlap between evaluation strategies. Manual human-in-the-loop evaluation is still the central strategy, but its co-occurrence with other methods is slightly lower in the all-data subset compared to deployed agents (Figure 10b), indicating that autonomous evaluation methods have more complementary roles in in deployed agents.

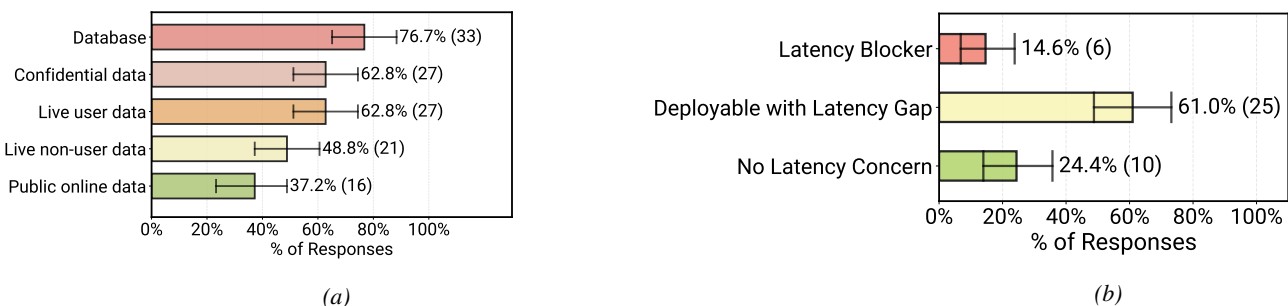

*(a)*                              *(b)*

*Figure 24.* [All Data] The figures correspond to Figure 12a (a) and Figure 12b (b) in the main text but include all survey data. (a) Types and modes of data ingestion and handling in all agent ($N = 43$). The distribution of data sources and handling methods did not change substantially when moving from deployed agents only to all survey data. (b) Degree to which latency causes problems for all agent systems ($N = 46$). The distribution of problematic latency did not change substantially when moving from deployed agents only to all survey data, suggesting latency is not a primary deployment blocker across the development lifecycle.

Figure 24 corresponds to Figure 12a and Figure 12b in the main text but reports statistics across all survey responses. Figure 24 shows distribution of data sources for agents which is remarkably stable when moving from deployed agents only ( Figure 12a) to [All Data] in Figure 24. This possibly indicates that the underlying data plumbing for agents is largely shared across lifecycle stages: teams tend to set up similar ingestion and handling pipelines during prototyping that then carry through to production with incremental hardening.

In addition, Figure 24b reports how often latency is described as a problem across all systems. The distribution changes only modestly when compared to the deployed-only subset (Figure 12b), and we again see that latency is not the dominant blocker for most agent deployments. This supports our broader conclusion that agents are currently concentrated in latency-relaxed settings where quality and correctness dominate over strict real-time responsiveness.

Finally, Figure 25 mirrors Figure 13 in the main text but includes all survey data. The overarching trend remains the same: growth is heavily concentrated in non-textual modalities, pointing towards increasingly multimodal agentic systems. Interestingly, the emphasis on future support for non-text modalities is even stronger than in the deployed-only subset, indicating that experimental and research agents are pushing more aggressively into multimodal directions (e.g., image, audio, and structured data) that may not yet have reached stable production deployment.

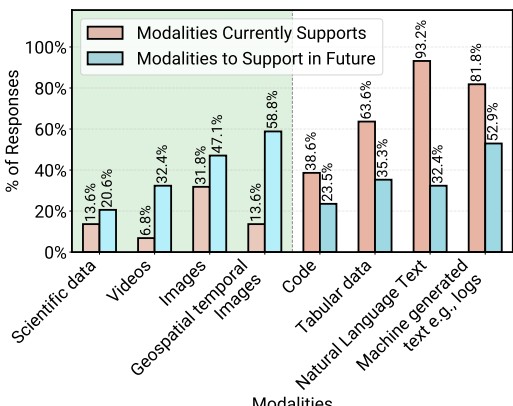

*Figure 25.* [All Data] Data modalities already supported (red) versus modalities planned for future support (blue) across all agents (corresponding to Figure 13 but for [All Data] . The trend of growth being heavily concentrated in non-textual modalities remains consistent, pointing toward increasingly multimodal agent systems. However, comparing this figure with the deployed-only subset shows that the full survey data in Figure 13 places an even stronger focus on non-textual modalities for future support. ($N$=44)

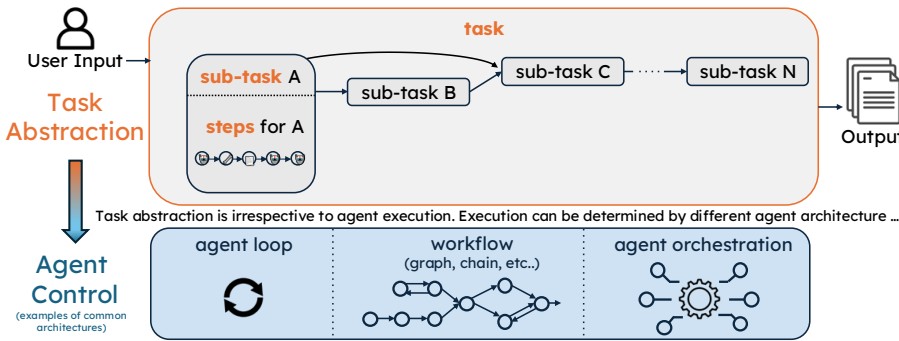

*Figure 26.* Conceptual visualization of our terminologies used in Section 5 and how it maps to the survey definition.

# E. Terminology

To ensure clarity and consistency, we established a hierarchical taxonomy for agent execution. Figure 26 provides a conceptual visualization of the key terminologies e.g., Task, Subtask, and Steps, as they are defined in our survey and applied throughout the paper. This mapping illustrates the relationship between high-level user goals and the granular autonomous actions taken by the agent.

# F. Literature Review Scope and Categorization

Table 4 summarizes representative prior work referenced in our literature review, organized according to the type of evidence each body of work provides about AI agents. We group works into three broad categories based on their methodological focus and the nature of empirical grounding they offer.

**Commercial/Industry**. These studies consist of practitioner and industry reports that describe agentic systems deployed in organizational settings. These accounts often provide high-level insights into trends, adoption patterns, or organizational impact, but typically lack methodological transparency, reproducible evaluation protocols, or detailed technical characterization of deployed systems.

**Research Surveys**. This group of work includes academic survey papers that synthesize existing literature on AI agents, multi-agent systems, or related paradigms. While these surveys offer valuable conceptual frameworks and comprehensive overviews of prior research, they generally do not incorporate primary data from production deployments, nor do they empirically characterize how agents are built, evaluated, and operated in real-world settings.

**Application/Demonstration**. This category comprises works that introduce specific agentic systems, frameworks, or

*Table 4.* Representative works from our literature review, organized into three categories based on evidence type: (1) high-level industry accounts with limited methodological detail, (2) academic surveys synthesizing existing research without primary production data, and (3) application-specific demonstrations lacking field-level characterization.

| Category | References |
|---|---|
| **Commercial/Industry** | Capgemini Research Institute (2025), Mic (2025), PagerDuty (2025) |
| **Research Surveys** | Chandra et al. (2025), Chen et al. (2024), Cheng et al. (2024), Dam et al. (2024), Du et al. (2025), Guo et al. (2024) He et al. (2024), Krishnan (2025), Liu et al. (2025), Luo et al. (2025), Ma et al. (2025), Masterman et al. (2024), Mohammadi et al. (2025), Piccialli et al. (2025), Plaat et al. (2025), Shang et al. (2025), Sun et al. (2025), Tran et al. (2025), Wang et al. (2024b), Xi et al. (2025), Yehudai et al. (2025) |
| **Application/Demonstration** | Abramovich et al. (2025), Anthropic Engineering Team (2025), Block, Inc. (2025), Chennabasappa et al. (2025), Cline Bot Inc. (2025), Gemini CLI Maintainers (2025), Gottweis et al. (2025), Jha et al. (2025), Kon et al. (2025), Wang et al. (2024c), Vahedian Movahed & Martin (2025), Schmucker et al. (2024), Shen et al. (2023), Park et al. (2023), Park et al. (2024), Parmar et al. (2025), Prabhakar et al. (2025), Singh et al. (2025), Teo et al. (2025), Wang et al. (2024a), Yang et al. (2024), Zhang et al. (2025) |

application-level demonstrations. These studies often showcase compelling capabilities or task-specific successes, but are typically limited to single systems or controlled scenarios and do not aim to provide field-level insights across organizations, deployment contexts, or operational constraints.

Together, these categories highlight a gap between existing conceptual, demonstrative, and industry-facing accounts and empirically grounded analyses of AI agents in production. Our study complements prior work by providing systematic, cross-organizational evidence on the technical methods, architectural patterns, and operational practices underlying deployed agentic systems.

## G. Survey Questionnaire Details

We crafted the questionnaire iteratively, refining it through early practitioner discussions. To facilitate broad participation, we limited the length, technical-depth, and disclosure-depth necessary to complete the questionnaire. All questions were optional and participation was entirely voluntary. Proceeding beyond certain questions required an answer or input however. For example, no participant could proceed without confirming they had read and understood the survey participation consent statement. Figure 28 and Figure 29 shows the control-flows of the Core and Additional sections of the questionnaire respectively. Further, all questions are intended for practitioners building AI Agents. Respondents who reported making technical contributions to more than 1 system (Table G.3 N1) were asked to focus all subsequent responses on 1 system of their choice (Acknowledgment G.1). Those who reported that they did not contribute to any systems they personally refer to as "AI agents" or "Agentic AI" systems were offered several options, such as commenting on terminology or their reason for starting the questionnaire, before being offered an early exit.

### G.1. Survey Acknowledgments

All participants who stated they worked with more than 1 AI agent or agentic AI system were required to acknowledge the following statement before proceeding further.

### Acknowledgment 1

To ensure consistency in your responses, please choose one agentic system to focus on throughout this survey. All your answers should relate to this same system. You may choose:

- A system you are most familiar with, or
- The system that is most developed among those you know — meaning closest to production with the most users.

Please feel very, very welcome to submit additional survey responses for each system you are familiar with.

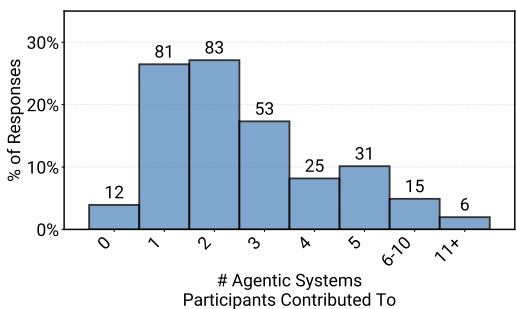

*Figure 27.* Distribution of the number of Agentic AI systems that participants reported contributing to ($N=306$).

## G.2. Survey Questions

We designed the questions to avoid response priming and facilitate downstream quantitative analysis, resulting in the question-type distribution shown in Table 5.

## G.3. Participant Contribution Distribution

As shown in Figure 27, this plot presents the distribution of how many Agentic AI systems participants reported contributing to. Across the $N=306$ systems represented in the survey, most respondents contributed to only a single system, with fewer individuals reporting involvement in multiple systems. This distribution highlights the breadth of participation across distinct agent deployments rather than concentrated contributions from a small subset of practitioners.

The exact survey questions and response choices are shown in the following sections.

| Question Type | Frequency |
|---|---|
| MCSA | 24 |
| Free-Text | 8 |
| MCMA | 7 |
| Rank-order | 5 |
| Numerical Input | 3 |
| **Total Questions** | **47** |

*Table 5.* Distribution of Question Types in Survey. Abbreviations: MCSA (Multiple-Choice Single-Answer), MCMA (Multiple-Choice Multiple-Answer).

## Core Questions

| ID | Question | Response Choices |
|---|---|---|
| QN1 | How many systems do you contribute to that you would personally describe as "AI agent(s)", "agentic", or "assistant(s)"? Example Answer: 2. (Required) | **Numerical Input**
— |
| QN1.1 | Do your colleagues or stakeholders call the systems you work on agentic, and if so, would you be willing to answer additional questions about the one with which you are most familiar? Only shown if answer for QN1 is = 0 | **MCSA**
Yes and yes
Yes and no, end questionnaire
No and no, end questionnaire |
| QN2 | May we contact you to learn more about your agentic/assistant systems? If so, please provide contact information. It will not be shared beyond the collaborators of this study or used for any other purpose. The question can be postponed after QN14. | **Free-Text**
— |
| QN3 | With respect to the system you chose, are references available (code repositories, blogs, publications, training data, evaluation data, or benchmark links)? If so, please provide links. | **Free-Text**
— |
| QN4 | List as many keywords as you can think of describing the domains in which the target problem (opportunity) arises. | **Free-Text**
— |
| QN5 | Which of the following best describes the status of the agentic/assistant system on which you chose to focus your responses? | **MCSA**
**In active production** – Fully deployed and used by target end-users in a live operational environment
**Pilot or limited deployment** – Deployed to a controlled group of users or environments for evaluation, phased rollout, or safety/security
**Undergoing enterprise-grade development, not yet in pilot or production** – Actively being built, tested, or integrated, but not yet piloted or deployed
**Prototype for potential development** – A functional early version intended to (ideally) evolve into a production system
**Retired or sunset** – Previously in use or prototyping but now decommissioned, cancelled, or replaced
**Research or education artifact** – Experimental or demonstrative, never intended for production use
**Unknown** – The status is unclear or the question is not understood |

| ID | Question | Response Choices |
|---|---|---|
| QN5.1 | Approximately how many target end users are actively using this production agentic system daily on average? Only show if QN5 answered. | **MCSA**
0, 1-10, 10-49, 50-199, 200-499, 500-999, 1,000-9,999, 10,000-99,999, 100,000-999,999, 1,000,000 or more users, Not sure |
| QN5.2 | Approximately, how many tasks from target end-users is the system processing per day on average? Only show if QN5 answered. | **Free-Text**
— |
| QN6 | Which of the following best describes your primary role with respect to this agentic system? | **MCSA**
**Oversight & Strategy** — Executive or Senior Leadership (e.g., CTO, VP, Director), Product or Program Manager, Project Manager or Scrum Master
**Industry Development, Engineering, Research** — Basic or Advanced, Software Developer / Engineer, Machine Learning Engineer, Data Scientist or Analyst, Researcher or Scientist
**Academic Research & Engineering** — Scientists, Students, Life Long Learners
**Operations & Infrastructure** — MLOps / DevOps / Platform Engineer, System Administrator or IT Support
**Quality & User Experience** — Quality Assurance / Test Engineer, UX/UI Designer or Human Factors Specialist
**Communication & Learning** — Technical Writer or Documentation Specialist, Educator or Trainer, Student or Intern |
| QN7 | What are the ultimate target gains of enabling/deploying the system? Select the highest priority option(s). (Skip the question if you do not know.) | **MCMA**
**Workforce adaptation:** reducing human expertise-levels or training generally required for the tasks
**Removing cross-domain, interdisciplinary knowledge requirements,** skills or training requirements
**Increasing Productivity/Efficiency:** increasing speed of task completion over the previous human/automated system
**Replacing time-consuming, low-skill, low-attention tasks** with automation
**Mitigating Risk:** reducing otherwise high or highly variable risk or uncertainty
**Decreasing human hours required** regardless of skills, task complexity, workforce expectations
**Mitigating Failure/Loss:** Decreasing time-to-intervention (security breach, system failure, customer loss)
**Increasing user engagement** and/or increasing service quality
**Enabling completion of tasks not possible** with the previous human/automated system |
| QN7.1 | Order the selected options according to their respective priority level. Only shown if QN7 Answered. | **Rank-order**
Answers from QN7 |

| ID | Question | Response Choices |
|---|---|---|
| QN8 | Who (what) are the primary direct users or consumers of the agentic/assistant system? (Select one.) | **MCSA**
**Other AI Agent(s)**
**Other NON-agentic software systems, tools, services**
**Humans operating INSIDE organizational boundaries** (e.g. employees operating inside a company and not their external customers)
**Human customers, general audience, operating OUTSIDE the org** authoring the agentic AI / assistant system |
| QN9 | Referring to the previously selected description of direct users or consumers as "user" (human or non-human), what does the system require from users in terms of behavior(s), interaction(s), or role(s)? To the agent/assistant, the target end-user is an… | **MCMA**
**Operator** (user initiates tasks, provides guidance, and determines a task is finished)
**Approver** (without necessarily providing guidance to reach the solution, user approves a solutions generated by the agentic system)
**Observer** (user passively observes the agentic system is operating as expected)
**Optimizer** (user actively intervenes to provide correction or improve performance without necessarily being an Operator or Approver) |
| QN9.1 | Please sort the selected options with (1) as the primary intended role, (2) as a secondary role, and so on. Press and drag an option to sort. Only shown if QN9 Answered. | **Rank-order**
— |
| QN10 | Using the previous definition of "user", how many steps or cycles can execute autonomously until user input is required? | **MCSA**
Four or fewer; Ten or fewer; Tens; Hundreds; Thousands; Millions; More; There is no limit, potentially infinitely many steps could execute without user input or intervention |
| QN11 | What determines how many steps or cycles can execute before a user's input is required? | **MCSA**
Problem complexity
Non-determinism in the agentic planning or decision-making
Preset limits e.g. in the configuration, parameters, or defaults
I do not know, or I have not measured |
| QN12 | How are each agent's system prompts (instructions) constructed? | **MCSA**
By hand, hard coded strings or templates e.g. LangChain templates
Using semi-automatic prompt engineering or optimization e.g. DSPy
Combination of manual and LLM or AI agents prompt creation and refinement
Fully autonomously by agents
Using libraries or templates predefined by others e.g. open-source
I don't know |
| QN13 | What is the average (typical) estimated instruction length per agent in words or tokens? (Skip if you do not know.) | **Numerical Input**
— |

*Continued on next page*

| ID | Question | Response Choices |
|---|---|---|
| QN14 | What is the maximum allowable end-to-end result latency for the agentic/assistant system? | **MCSA** 
 I don't know 
 No limit set yet, still in exploratory phase, not a latency-critical system 
 Microseconds 
 Subsecond 
 Seconds 
 Minutes 
 Hours 
 1–4 days 
 Weeks 
 Months 
 More |
| QN2 | May we contact you or your colleagues to learn more about your agentic/assistant systems? If so, please provide contact information. It will not be shared beyond the collaborators of this study or used for any other purpose. Only shown if question was postponed. | **Free-Text** 
 — |
| SEP | This is the end of the core questions. Would you like to answer further questions? | **MCSA** 
 Yes; No |
| EOS1 | End of survey – any last comments or feedback can be shared here. Only shows if SEP is No. | **Free-Text** 
 — |

## Additional Questions

| ID | Question | Response Choices | Additional Context |
|---|---|---|---|
| QO1 | Select any/all of the following methods currently integrated into your system that give you confidence your agentic/assistant system is consistently producing high quality outputs, whatever "high quality" means in your context. | MCMA
**Manual (Human-in-the-loop):**
Expert Review
Manual Citation Verification
Crowdsourced Evaluation
Red Teaming
**Automated Not-Model-Based Cross-Referencing:**
External Fact-Checking
Knowledge Graph Validation
Automated Citation Verification
**Automated Model- and Estimation-Based Methods:**
Self-Consistency Checks
Internal Confidence Estimation
Critique Models
Red Teaming using models
Cross-Model Validation
LLM-as-a-Judge
**Automated Rule-Based Methods:**
Grammar and Syntax Checks
Domain-Specific Rules
**Other:**
None of the above/below
I am not confident the system consistently produces high quality output yet | Hover/click on the "i" for examples, definitions and descriptions.

**Definitions:**
Expert Review: Involve human specialists to validate content in high-stakes or sensitive domains
Manual Citation Verification: User ensures cited sources are accurate and actually support the generated claims
Crowdsourced Evaluation: Collect feedback from diverse human reviewers to assess quality and usefulness
Red Teaming: Humans manually test robustness by probing with adversarial or misleading prompts or actions to expose weakness
External Fact-Checking: Retrieve supporting evidence from trusted sources and validate claims e.g. using retrieval-augmented generation (RAG)
Knowledge Graph Validation: Cross-check facts against structured data like Wikidata or domain-specific ontologies
Automated Citation Verification: Ensure cited sources are accurate and actually support the generated claims
Self-Consistency Checks: Generate multiple answers and compare for consistency, use majority voting to select the most common answer, and/or apply chain-of-thought reasoning to ensure logical consistency
Internal Confidence Estimation: Score answers using log probabilities and/or estimate uncertainty with dropout or ensemble methods
Critique Models: Use separate models to evaluate factuality, coherence, and overall quality of the output
Red Teaming using models: Test robustness by probing with adversarial or misleading prompts to expose weakness
Cross-Model Validation: Compare outputs from different AI models to identify consensus or discrepancies; Use Zero-Shot Critics, unrelated models to critique outputs without prior task-specific training
Grammar and Syntax Checks: verify grammatical correctness and linguistic clarity with or without NLP models
Domain-Specific Rules: Apply expert-defined rules for accuracy, tailored to specific fields like business, medicine, law or finance |

| ID | Question | Response Choices | Additional Context |
|---|---|---|---|
| QO2 | Sort the following categories from most to least important for ongoing development and production deployment. Press and drag an option to sort. | **Rank-order** **Data and Model Integrity:** Data Quality and Availability Model Drift and Concept Drift Versioning and Reproducibility **Core Technical Performance:** Robustness and Reliability Scalability Real-Time Responsiveness Resource Constraints **System Integration and Validation:** Integration with Legacy Systems Testing and Validation Security and Adversarial Robustness **Transparency and Governance:** Explainability and Interpretability Bias and fairness Accountability and Responsibility **Compliance and User Trust:** Privacy and Data Protection User Trust and Adoption Regulatory Compliance | Hover/click on the "i" for examples and definitions. **Definitions:** Data Quality and Availability: Accessing clean, timely, and relevant data for decision-making Model Drift and Concept Drift: Adapting to changes in data distributions and task definitions Versioning and Reproducibility: Tracking models, data, and configurations for auditability Robustness and Reliability: Ensuring consistent, correct behavior in diverse and unpredictable environments Scalability: Supporting growth in users, data, and tasks without performance degradation Real-Time Responsiveness: Meeting latency and timing requirements in dynamic contexts Resource Constraints: Managing compute, memory, and energy efficiently Integration with Legacy Systems: Seamlessly connecting with existing infrastructure and APIs Testing and Validation: Simulating and verifying agent behavior before deployment Security and Adversarial Robustness: Defending against manipulation and exploitation Explainability and Interpretability: Making decisions understandable to humans Bias and fairness: Preventing discriminatory or unjust outcomes Accountability and Responsibility: Clarifying who is liable for agentic decisions Privacy and Data Protection: Ensuring compliance with data regulations (e.g. GDPR) User Trust and Adoption: Building confidence through transparency and reliability Regulatory Compliance: Meeting legal standards for autonomy, safety, and transparency |
| QO3 | Have you compared your agentic/assistant solution to a non-agentic solution, which of the following statements is most accurate? | **MCSA** Yes; No, alternatives might exist but I have NOT formally compared them; No, alternates DO NOT exist, my system provides truly novel functionality | — |
| QO3.1 | If it were entirely up to you, would you choose the agentic solution over the alternatives? Only shown if answer for QO3 is Yes | **MCSA** Yes No | — |

| ID | Question | Response Choices | Additional Context |
|---|---|---|---|
| QO3.1.1 | You answered "Yes" to "If it were entirely up to you, would you choose the agentic solution over the alternatives?" Why, which of the following functional improvements does the new system offer compared to the previous solution? (Select all that apply.). Only shown if answer to QO3.1 is Yes | **MCMA** 
 Increased automation or reduced manual effort 
 Enhanced user interface or usability 
 Scalability or support for more users 
 ONLY Ease of software design, maintenance, model use or integration 
 Improved performance or speed 
 For non-technical reasons ONLY, e.g. marketing/advertising, strategic planning, ... 
 Better integration with other systems 
 Improved data accuracy or consistency 
 Enhanced security or compliance 
 Expanded features or capabilities 
 None of the above | — |
| QO3.1.1 | Order all selected options according to their respective priority level. - Improved performance or speed (Options from QO3.1.1). Only shown if answer to QO3.1.1 has answers | **Rank-order** 
 Answers from QO3.1.1 | — |
| QO4 | Thinking about the state of the system you chose to focus on, in your personal opinion, would you call it an "assistant" but not an AI Agent or agentic? | **MCSA** 
 Either/Both; Assistant and NOT Agent or agentic | — |
| QO4.1 | Why do you refer to your system as AI agents or agentic? What makes it agentic in your opinion? Only shown if answer to QO4 is Either/Both | **Free-Text** 
 — | — |

| ID | Question | Response Choices | Additional Context |
|---|---|---|---|
| QO4.2 | Why do you refer to your systems or an "assistant rather than an "agent" or "agentic"? What makes it an assistant rather than an agentic system in your opinion? You may go back and change your previous answer if you use any of these terms to describe your system. Only shown if answer to QO4 is Assistant and NOT Agent or agentic | **Free-Text** — | — |
| QO5 | What is the minimum level of expertise or training (knowledge and skills) expected from typical end-users? (Select the best option.) | **MCSA** **Highly skilled professionals** — Advanced education and specialized knowledge (e.g., engineers, scientists, doctors); capable of complex tasks like coding, diagnostics, or system design **Extensive domain experience** — Deep practical knowledge from years of experience; having organizational or domain knowledge, skilled in nuanced tasks like troubleshooting or decision-making **General education** — High school level education with basic digital skills (e.g., using email, spreadsheets, or web apps) and standard subject matter knowledge **Minimal expertise required** — Little to no formal education; able to follow simple instructions or perform basic tasks (e.g., tapping buttons, entering data) **Not sure** | — |

| ID | Question | Response Choices | Additional Context |
|----|----------|------------------|--------------------|
| QO6 | How would you rate the return on investment (ROI) of this agentic system relative to its total cost of development, operation, infrastructure and all other costs? (Please select the option that best reflects your assessment.) Only shown if answer to QN5 is "In active production – Fully deployed and used by target end-users in a live operational environment" | **MCSA** Exceptionally high ROI (ROI greater than 150%) High ROI (ROI between 125%–150%) Acceptable ROI (ROI between 90%–124%) Low ROI (ROI between 60%–89%) Poor ROI (ROI less than 60%) | — |
| QO7 | How much do each agent's system prompt (instruction) lengths vary? (Skip if you do not know.) | **MCSA** Tens of tokens Hundreds of tokens Thousands of tokens Ten of Thousands of tokens More | — |
| QO8 | Compared to the target latency for the system's result turn-around, how problematic is the actual latency? (Select one.) | **MCSA** I don't know I don't understand the question Not problematic at all, the actual latency is better than expected and not a problem for deployment Marginally problematic, but good enough for deployment Very problematic, the system can't be deployed without addressing the gap, or the highest priority post-deployment will be bringing down the latency | — |
| QO9 | What is the maximum target latency for determining a single action, step, or response? (Select one.) | **MCSA** Microseconds; Subsecond; 1 to 10 seconds; 10 to 60 seconds; A few minutes; More; Less; I don't know; I don't understand the question | — |

*Continued on next page*

| ID | Question | Response Choices | Additional Context |
|---|---|---|---|
| QO10 | What is the maximum number of distinct models used together to solve a single logical task? Skip if you do not know; estimates are fine. | **Numerical Input** — | — |
| QO11 | Are inference time scaling techniques used in your system? | **MCSA** Yes; 0, No inference time scaling is used; I don't know | Help: For the purposes of this question, "inference time scaling" a.k.a. "test time compute" refers to a family of techniques that call models multiple times or use a collection of models together, in place of a single model call and without modifying the weights or retraining any models, to answer a single question, choose a single next step, action, tool, etc... |
| QO11.1 | If inference time scaling techniques are used, approximately how many model calls are made per user query or task? Only shown if answer for QO11 is Yes | **MCSA** Tens Hundreds Thousands Tens of thousands Hundreds of thousands Millions More | — |
| QO12 | What data modalities does the system process now (today)? | **MCMA** Natural Language Text Tabular data Software or machine generated text including system logs, events, etc. Code Images Videos Image sequences or batches with additional channels or metadata e.g. geospatial, NMR scans Scientific data not listed above | — |
| QO13 | What data modalities will or should the system process in future (that it does NOT process now)? | **MCMA** Same as QO12 | — |

| ID | Question | Response Choices | Additional Context |
|---|---|---|---|
| QO13.1 | You have selected the following option(s) for the question "What data modalities will or should the system process in future (that it does NOT process now)?": What are the barriers to processing them now? Sort the options from most to least important; press and drag an option to sort. Only shown if QO13 is answered. If you don't know, skip the question. - No barriers, just a matter of development time | **Rank-order** 
 Answers selected from QO13 | — |
| QO14 | Which describes the data handling functions the system(s) perform? Select all that apply and use the "QOther" box to detail. | **MCMA** 
 Ingests direct user input in real-time 
 Ingests other real-time information as well as direct user input 
 Ingests information from other systems, e.g. databases, at most indirectly from end-users 
 Retrieves persistent data from public external sources (e.g. the web) 
 Retrieves non-public, confidential, or otherwise federated data 
 Other | — |
| QO15 | Are you using an openly (commercially or non-commerically) available agent-focused programming framework (e.g. LangChain, CrewAI, Autogen,...) to implement your system? | **MCSA** 
 Yes 
 No, only an in-house not openly available solution, or only standard programming languages and tools e.g. Python, Java (not-agent-focused) 
 I don't know | — |

| ID | Question | Response Choices | Additional Context |
|---|---|---|---|
| QO16 | From experimental observations, which of the following supports most of your assistant/agentic system functionality or design? Only shown if answer to QO15 is Yes | **MCSA**
OpenAI Swarm
CrewAI
LangChain or LangGraph
BeeAI
Autogen or AG2
LlammaIndex
Other: | — |
| QO17 | How long has it been since active prototyping or initial development started on this agentic/assistant system? | **MCSA**
Less than 3 months; 3–6 months; 6–12 months; 1–2 years; 2–5 years; More than 5 years; I don't know or prefer not to say | — |
| QO18 | How long have you been working on this agentic/assistant system? | **MCSA**
Same as QO17 | — |
| EOS2 | Thank you, this completes our questionnaire. Go back to change any of your answers, or click END to finalize them and exit. Any last comments or feedback can be shared here. | **Free-Text**
— | — |

**Analysis Aspects Overview**   Given the diversity of agent applications and architectures, we pinpointed 17 common aspects for analysis, summarized below. Regarding respective data collection, see the interview guide (Appendix C.3) and example survey question references (e.g., *QN13, QO7* with full text in Appendix G).

1. **Prompt instruction length** (averages, variance, distributional properties) – e.g., *QN13, QO7*

2. **Prompt instruction construction methods** (automated, semi-automated, manual, hybrid) – e.g., *QN12*

3. **Model selection** (open- versus closed, post-training)

4. **Number of models used** (single-model vs. multi-model architectures) – e.g., *QO10*

5. **Use of inference-time techniques** (e.g., RAG, caching, routing, speculative decoding, batching) – e.g., *QO11, QO11.1*

6. **Agent architecture: control flow, etc.** (branching, looping, tool invocation, degree of autonomy) – e.g., *QN9, QN10, QN11*

7. **Agent dependencies** (other agents, external software, human oversight, human expertise, data sources) – e.g., *QN9, QN10, QN11, QO5*

8. **Data sources** – e.g., *QO12*

9. **Data modalities** – e.g., *QO13*

10. **Programming frameworks** – e.g., *QO15, QO16*

11. **Metrics** – e.g., *QN7*

12. **Evaluation and verification methods** (trustworthiness, accuracy, safety) – e.g., *QO1, QO3*

13. **Baseline evaluations /** (human, agents, or non-agent systems) – e.g., *QO3.1.1*

14. **System throughput and latency** – e.g., *QN5.2, QN14, QO8, QO9*

15. **Technical challenges** (current limitations and ongoing development priorities) – e.g., *QO2*

16. **Application domain** (domain(s), tasks, and deployment context) – e.g., *QN4, QN3, QN7, QN8*

17. **System stage** (prototype, pilot, partial deployment, production) – e.g., *QN3, QO17, QN5, QN5.1, QN5.2*

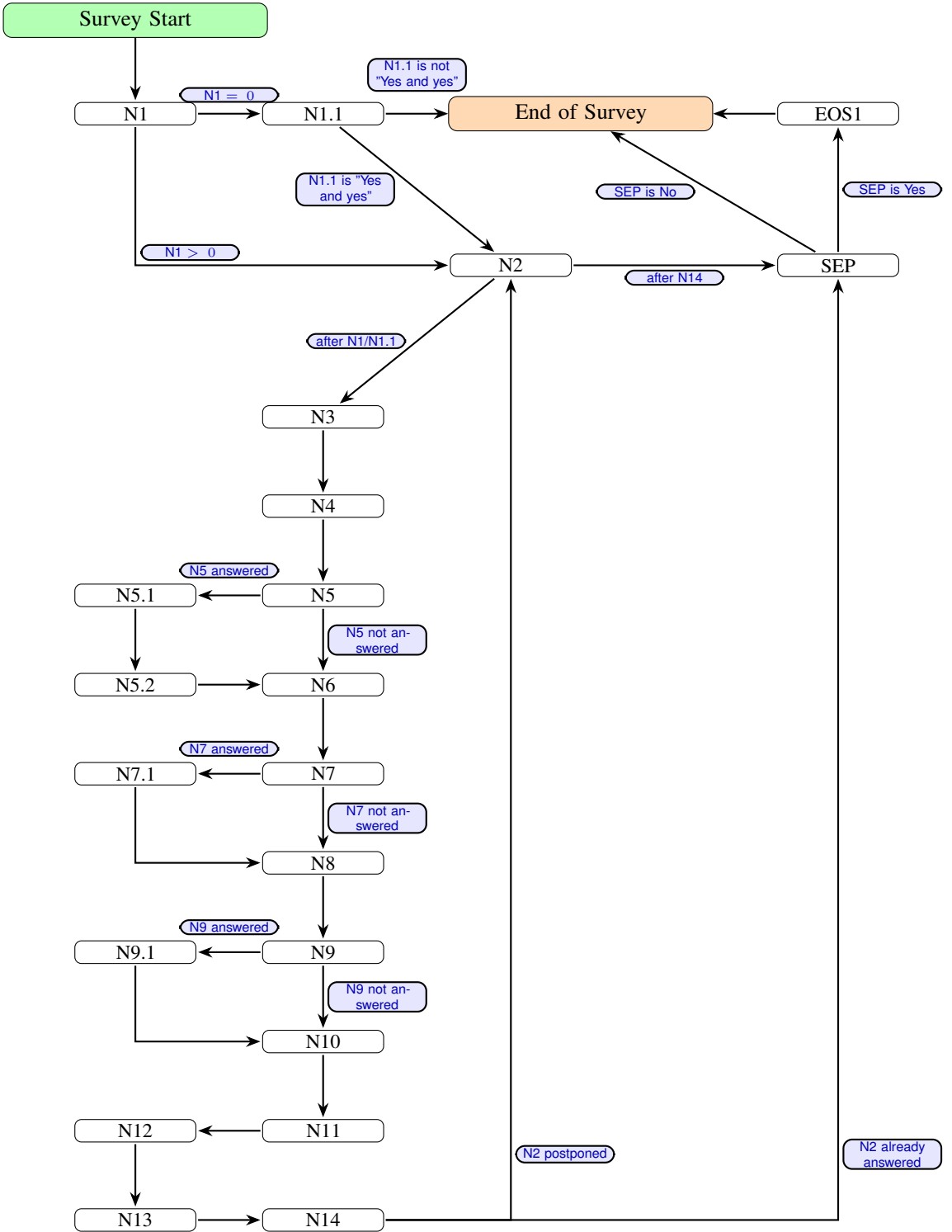

*Figure 28.* Survey flow: core questions. The exact text for each question can be found Table G.3 e.g. N1 is QN1 in Table G.3.

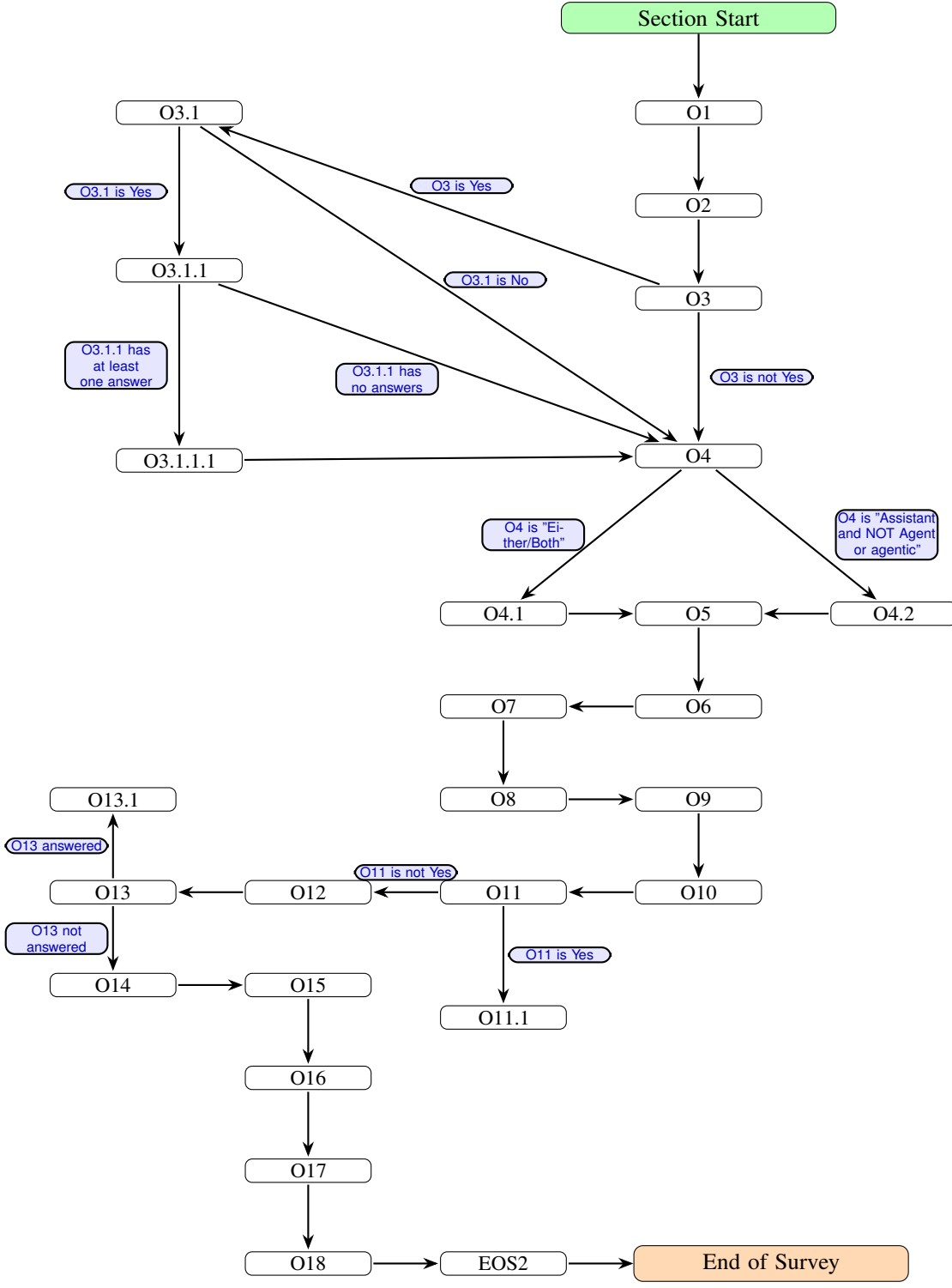

*Figure 29.* Survey flow: additional questions. . The exact text for each question can be found Table G.3 e.g. O1 is QO1 in Table G.3.

