# OpenReview forum: "Measuring Agents in Production"
_ICML.cc/2026/Conference — ICML 2026 spotlight_

### Official Review · Reviewer_xrdf · 2026-03-12

**Soundness:** 3
**Presentation:** 3
**Significance:** 4
**Originality:** 3
**Overall Recommendation:** 5
**Confidence:** 4

**Summary:**

This paper is an empirical study of how LLM agents are actually built and deployed in real-world settings. Rather than proposing a new algorithm, it aims to document current production practices using first-hand practitioner data. The authors combine 20 in-depth case studies with a survey of 306 practitioners, and focus their main analysis on 86 production or pilot systems across 26 domains.

The conclusions drawn from this work are meaningful. The paper suggests that teams mostly use off-the-shelf models, manual prompting, and static workflows, and that only a small fraction use agent frameworks. The paper also finds that human oversight is central. Most teams rely on human-in-the-loop evaluation, while many skip formal benchmarks and instead use A/B testing or expert feedback. The biggest deployment issue is reliability, followed by evaluation and security, so successful systems often depend more on system-level safeguards and bounded workflows than on highly autonomous agent behavior.

Overall, I think this paper makes a strong contribution. Some of its conclusions are counterintuitive. For example, it suggests that real production agents are much simpler than research papers often imply.

**Compliance With Llm Reviewing Policy:**

Affirmed.

**Final Justification:**

The rebuttal fully addresses my concerns. I recommend acceptance.

**Key Questions For Authors:**

N/A

**Limitations:**

See weakness.

**Strengths And Weaknesses:**

**Strenghth:**
1. Important problem setting: The paper studies a question the community cares about but rarely measures systematically: how LLM agents are actually deployed in production. That makes the paper valuable even without a new algorithm.
2. Clear and useful findings: The paper delivers several concrete takeaways: production agents tend to use off-the-shelf models, manual prompting, static workflows, and frequent human oversight; reliability remains the top bottleneck. These findings are actionable and relevant to real world agentic system designing.
3. Great writing: The paper is well written and easy to follow. The presentation is rigorous and professional.

**Weakness:**
1. Some findings may be unsurprising: Claims such as "Practitioners primarily build agents for productivity gains through automation" are sensible.  This observation would likely remain true even if “agents” were replaced with “AI systems” more broadly.
2. Most of these conclusions are drawn from interviews and surveys. A stronger study would require direct auditing of deployed systems, since the interviewed participants may not fully represent the broader teams responsible for designing, deploying, and maintaining these agents.

But overall, I think this paper is meaningful and provides valuable guidance to the broader community.

---

> ### Author Rebuttal · Authors · 2026-03-30
>
> Thank you for your valuable time, helpful review, and strong assessment. We are grateful for your recognition that this paper studies an **important problem setting** the community cares about, delivers **clear and useful findings** on real production practice, and offers **valuable guidance to the broader community** through a well-written and rigorous presentation. We are especially glad that you found the paper’s empirical contribution meaningful even without proposing a new algorithm.
>
> Since submission, we continue to analyze the study corpus in more depth and extract new insights. These additional analyses reinforce the same central pattern: deployed teams heavily rely on off the shelf models, emphasis on control via simple design choices, and human oversight, with reliability remaining the main bottleneck. In a fast moving area, we believe having a grounded baseline of how the systems work becomes more valuable, because meaningful research requires a deep understanding of the real-world problem space. *MAP is a start of this line of work*, and we are continuing extending it to broader coverage with new datapoints and insights for the community.
>
>
> ***
> # Findings
> > W1: potentially unsurprising findings, such as the top reason behind why agent systems are built is productivity gain
>
> We first thank the reviewer for acknowledging that many of the paper’s findings are insightful and counterintuitive.
>
> We also agree that some findings, such as productivity gain, may appear intuitive (less surprising) at a high level. This follows from our goal: we aim to document production agent practice in a broad and neutral way. For that reason, we report both **confirmatory findings** that establish a baseline or knowledge for the field and **counterintuitive/less expected findings** that challenge common research narratives, such as the prevalence of simple workflows, limited autonomy, and frequent human oversight.
>
> We also go beyond high level finding summaries by adding finer grained detail. For example, beyond showing that productivity is the primary motivation, we show that teams most often measure it through **end to end completion time**, not only through model centric metrics. We hope this level of detail helps and underlying metrics or reason behind design choices can help the community better understand the requirements and principles of agent systems in real-world settings.
>
>
> ***
>
> # Direct Auditing
> > W2: can enhance the study via direct auditing the deployed agent systems.
>
> We thank the reviewer for this suggestion. We agree that direct auditing would add another useful lens. At the same time, interviews and surveys are essential for answering the research questions in this paper, such as **why** teams choose bounded workflows, **how** they evaluate model changes, **which operational constraints** shape deployment, and **what is the top development challenge** block broader rollout. These are questions for which practitioner-centered evidence is especially valuable, whereas direct system auditing from a user perspective can be less well suited to providing this type of insight.
>
> Thus, we see direct auditing as a great complementary next step rather than an alternative to interviews and surveys for the scope of our study. In fact, our team is already actively pursuing this direction as a follow-up to MAP, and we hope to share those results with the community in future work.
>
> ***
>
>
> Thank you again for your valuable time and thoughtful feedback. Your comments help us sharpen both the framing and the methodological contribution. We hope these clarifications make the paper’s contribution clearer. If so, we would be grateful if you could reflect that in your updated assessment.
>
> Best,
>
> The Authors

---

> > ### Author Rebuttal · Reviewer_xrdf · 2026-04-03
> >
> > Thanks for the rebuttal. I will keep my score.

---

### Official Review · Reviewer_A99N · 2026-03-13

**Soundness:** 3
**Presentation:** 4
**Significance:** 4
**Originality:** 4
**Overall Recommendation:** 5
**Confidence:** 4

**Summary:**

This paper shares findings from a systematic study to understand how agents are used in production. 20 case studies were conducted from companies of varying sizes (start-up to mature), with 306 practitioners surveyed over 26 domains. The main findings are that agents tend to be used for productivity purposes, most use simple custom agent frameworks with existing closed-source models, most agent evaluation is done by humans instead of curated benchmarks, and the top challenges in deployment are mainly reliability. Many of the top choices of actual agents in production are contrary to popular directions in research, such as post-training, suggesting that there are many underexplored research topics for others to tackle.

**Compliance With Llm Reviewing Policy:**

Affirmed.

**Key Questions For Authors:**

I found this survey to be quite informative and believe it would be of great use to the machine learning community to inspire new research directions. I had a few questions regarding information that wasn't evident.

(1) Regarding operational constraints, it seems that multiple models are maintained in cases where a new model update will break previous behaviors, but how do the practitioners manage the comparison of multiple models when they are doing human-in-the-loop evaluation (I imagine this can quickly become cumbersome). What percentage of teams mentioned using practices like these?

(2) Do the teams that have low-latency requirements prefer to perform model weights tuning or do they use proprietary models?

(3) What is meant by "reliability" in this paper and how did other practitioners define it? From Section 7.1 it seems reliability refers to mitigating errors and practitioners do this through extensive human interventions but reliability has also been defined as consistent success over the same task in other contexts [1].

[1] τ-bench: A Benchmark for Tool-Agent-User Interaction in Real-World Domains.

**Limitations:**

yes

**Strengths And Weaknesses:**

$Soundness$: The description for how the study was designed is clearly explained and the limitations are transparently laid out (e.g. geographically concentrated in Americas, studies skewed towards authors' professional connections)

$Presentation$: The research questions are presented at the start and the paper is well-structured towards answering the questions and clearly presenting the findings

$Significance$: This paper provides very useful insight into how agents are actually deployed in the real-world which is crucial to understand what works in the wild and what researchers need to focus on to improve existing real-world methods.

$Originality$: This is one of the first surveys to gather information on engineering-level technical data on production systems from practitioners that build agents.

---

> ### Author Rebuttal · Authors · 2026-03-30
>
> Thank you for your valuable time and strong assessment. We appreciate your recognition that MAP is among the **first systematic studies** of how agents are built in production, that it provides **useful insight into how agents are actually deployed in the real world**, and that it can **inspire new research directions for the ML community**. We are especially glad that you find the paper well structured, informative, and significant.
>
> We appreciate that you found no major weaknesses and instead raised three insightful follow-up questions. We provide our responses to each point below, and your questions help us to improve the depth of our study which we will include in the next revision.
>
>
> ***
> # Agent evaluations
> > Q1: How do teams evaluate model upgrades if human-in-the-loop dominates the evaluation process?
>
> This is an insightful question. We conduct an additional pass over the interview data and find that two teams that explicitly report maintaining multiple models *due in part to model upgrades* also report using **hybrid evaluation**: curated golden answer benchmarks or pre built CI/CD regression tests together with subject matter expert review. In other words, human in the loop evaluation is often not used in isolation for model comparison in model upgrades.
>
> More broadly beyond model upgrade, human-in-the-loop evaluation is often used alongside other methods such as LLM-as-a-judge, rather than in isolation. We show the co-occurrence of evaluation methods in Figure 10B in Appendix B, and we will highlight this result more clearly in the main text.
>
> ***
>
> # Low-latency requirements correlation with weight tunning
> > Q2: Do teams with low-latency requirements prefer model weight tuning, or do they use proprietary models?
> >
> > **Action:** We will add the new post-survey analysis below to the main paper.
>
> We conduct an additional cross case pass on this question. Among the **4 interview cases** with explicit bounded latency requirements, **3 use weight tuning**, and the remaining **1 uses a proprietary model** together with system level techniques such as caching. This is a small sample, so we do not claim a broad prevalence result, but it does suggest that latency sensitive teams often adapt models or system design directly rather than relying only on frontier proprietary APIs. We will add this pattern as an observation to the paper.
>
> * Systems that fine-tune: *C14, C04, C02*
> * Systems that use proprietary models: *C05*
>
> ***
>
> # Definition of reliability
> > Q3: What is the definition of reliability?
> >
> > **Action:** We will add both the definition and the new analysis below to the main paper.
>
> We follow IEEE’s definition of reliability “Software reliability is the probability of failure-free software operation for a specified period of time in a specified environment.” [2]. This is also broadly consistent with the definition of repeated task success notion discussed in [1]. We will add this definition explicitly in the relevant sections and cite the corresponding literature.
>
> [1] τ-bench: A Benchmark for Tool-Agent-User Interaction in Real-World Domains.
>
> [2] IEEE Computer Society, IEEE Standard Glossary of Software Engineering Terminology, IEEE Std 610.12-1990, 1990.
>
>
> **Reliability failure Analysis**: in addition, we conducted a new pass over the interview data to identify more specific reliability challenges and failure modes once encountered by agent teams during systems development or runtime phase. Three patterns appear repeatedly:
>
> * Incomplete or missing evaluation leads teams to rely on human feedback and subject-matter-expert verification while building task-specific evaluation datasets and benchmarks from the ground up. (*C01a, C03a, C05a, C09a, C14a, C15a, C17a*)
> * Correctness failures increase with application complexity, especially when systems combine heterogeneous or multimodal data sources. Across these settings, teams often fall back to expert human verification. (*C03a, C06a, C07a, C10a, C14a*)
> * Legacy system integration, including existing security and compliance requirements, can limit functionality or narrow deployment scope. (*C03a, C04a, C10a, C15a, C16a, C16b*)
>
> We will report these finer grained categories in the main text.
>
> ***
>
> Thank you again for your thoughtful feedback and strong support. Your questions help us sharpen the analysis and make the contribution clearer. We hope these clarifications and additional analyses address your questions. If so, we would be grateful if you could reflect that in your updated assessment. Looking forward to hearing from you.
>
> Best,
>
> The Authors

---

> > ### Author Rebuttal · Reviewer_A99N · 2026-04-04
> >
> > Thank you for the response and considerations in adding the new findings to the paper. I will be maintaining my strong recommendation.

---

### Official Review · Reviewer_z4SD · 2026-03-16

**Soundness:** 3
**Presentation:** 3
**Significance:** 3
**Originality:** 2
**Overall Recommendation:** 5
**Confidence:** 3

**Summary:**

The paper presents a survey which characterises how (and why) agents in production are built and evaluated, with novel data sourced from interviews with real practitioners and developers across relevant domains. They also investigate causes of success and failure in production agents; finding, for example, that long horizon reliability remains a consistent failure mode of agents in practice, that most practitioners prefer simplicity over autonomy and in context learning over finetuning, and that developers largely rely on human evaluation (e.g. spot-checking, casual testing) rather than running benchmarks.

**Compliance With Llm Reviewing Policy:**

Affirmed.

**Final Justification:**

Most of my concerns were clarified or already addressed in the appendix. The new reliability failure mode analysis is a useful addition. Keeping at 5.

**Key Questions For Authors:**

- Can you detail how the noted limitations might distort any of the paper's findings, and what affect this has on the validity of the paper's claims?
- Do you have any insights as to what types of reliability failures affect practitioners the most heavily?

**Limitations:**

As mentioned above, the limitations section is quite concise, and would be helped by the addition of discussion surrounding some of the points above.

**Strengths And Weaknesses:**

Strengths:
- **The paper addresses a widely important issue** and presents genuinely novel insights and information sourced from interviews with deployers of production agents.
- **The paper presents its main research questions clearly,** making the contributions and results very easy to follow.
- **The key findings are novel, informative, and impactful:** Several of the key findings are interesting and, to me, unexpected - for example, I am surprised by how few practitioners formally benchmark their agents, and that the overwhelming majority of agents are used with a human in the loop, rather than within automated systems. Other findings are less surprising (agents increase productivity, most practitioners use frontier models and in context learning), but nonetheless are valuable to have validated in the scientific literature.
- **The paper has a large sample size (306 participants) with experts across a wide variety relevant domains.**
	- However, not that large after filtering - TODO: remove?
- **The the full survey data is made available, including survey questions and responses.** This will be valuable for future analysis, and makes the paper more transparent, and more easily reproducible.
- **A good range of related literature is presented**, and the authors clearly lay out where their contributions differ from it (this paper is focused on engineering details of production systems).
- **The paper is transparent about study design choices and motivations** - e.g. participant location, domain, interview setup.
- The survey design is strong: There is an emphasis on variety as well as quantity of sources, the semi-structured interviews - provide novelty and allow quantification and comparative analysis. The choice to use grounded theory as an analysis method is also appropriate, however *the details of how the grounded theory analysis was run are not reported in the paper*. For example: did you conduct the open code labelling with human participants, or automate it with a language model?
- **The paper is sufficiently statistically rigorous.**
- **Clear data visualisations** which complement but don't repeat the information in the text

Weaknesses:
- **The study covers a limited geographical area, and could have issues with locational and demographic bias.** The authors do not, as far as I can tell, report any group-level demographic data of the study participants (although understandably does not include individual-level demographic data for confidentiality reasons).
- **Limitations section is too concise:** Whilst the limitations section has good coverage of potential study biases which may affect the validity of their results, it is somewhat concise. More detail could be added about lower level limitations, for example the interview structures, and areas where their question choices may lack coverage.
- **The limitations section should discuss in more detail how limitations could systematically distort overall findings**, or specific findings.
	- For example, temporal bias: whilst the authors do mention potential temporal bias in the limitations section, its relevance is not highlighted as clearly it should be. The period of data collection spans April to November 2025, which was a period of extreme progress in agentic systems - with SOTA results on swebench, for example, increasing by 20+ pp from 50-60%, to 80% . This negatively affects the generality of the study and is not highlighted as clearly as it should be.
- **The paper could be improved by the inclusion of some qualitative examples detailing specific case studies and interview questions.**
- **The number of total systems and case studies is fairly small,** and the abstract makes the sample size sound more substantial on first glance by citing their 306 survey respondents. However, only 86 survey respondents remain after filtering for deployed systems, and there are only 20 individual total case studies. These sample size limitations should be highlighted more clearly in the limitations section or the abstract.
- **The paper mentions reliability as a core failure mode of agentic systems, but doesn't cover specific reliability failure modes in the main body.** There is some discussion of this in the appendix, but I think a) it would be a valuable addition to the main body of the paper to discuss more detailed insights about failure modes from the participant interviews, if space allows; and b) the challenge categories should be more fine grained - a wide range of issues fit under the "Core Technical Performance" category, making it hard to identify what specific robustness failures participants encounter most frequently.

---

> ### Author Rebuttal · Authors · 2026-03-30
>
> Thank you for your valuable time and detailed review. We appreciate your recognition that our paper addresses a **widely important problem**, offers **novel and impactful insights from real production practitioners**, uses a **large and diverse sample**, provides **open survey data**, and is supported by a **strong study design, clear presentation, and statistically rigorous analysis**. We are glad that you found both the findings and the paper structure easy to follow.
>
> Since submission, we continue to analyze the study corpus in more depth and extract new insights. These additional analyses reinforce the paper’s central findings: deployed teams prioritize reliability, control via simplicity, and human oversight. More broadly, we hope MAP helps the community towards understanding the production problem space deeply to guide new research. Please find our responses below.
>
> ***
> # Geographic context
> > W1: report group-level demographic data of the study participants
>
> We agree that group-level context should be easier to see. We already report this in Appendix C.2.2 and will reference it explicitly in the main methodology section. At the case level, our 20 sources span diverse geographies: **11 span 5-6 continents, 4 span 2-4 continents, and 5 are concentrated in one continent**; **13 span tens of countries, 1 spans hundreds of countries, and 5 are concentrated in one country**. We will make this easier to locate in the main paper.
>
> ***
> # Methodology details: limitations and grounded-theory
> > Q1, W2, W3: expand on potential biases, how they may affect findings, and clarify the grounded-theory analysis
>
> We agree and will expand this discussion in the revision.
>
> **Temporal coverage.** We agree that the April-November 2025 collection window sits in a fast-moving period. This can shift some observed distributions, such as how many steps teams allow before human intervention (agent autonomy). We will state this more clearly. At the same time, our additional post-submission analysis continues to support the same higher-level conclusion: production choices are still driven primarily by reliability. Thus, the goal and hope of MAP is that the ***higher-level design principles behind these specific distributions may remain relevant to support agent research for the broader research communities***.
>
> **Interview structure.** Because we use semi-structured interviews, some fine-grained design choices are discussed only in a subset of cases. This means rare patterns cannot be evaluated quantitatively, so we report them as qualitative examples.
>
> **Grounded theory.** We will also clarify the coding procedure in the paper: all open coding is done by humans, with at least **3 researchers per interview note**, and disagreements are resolved via peer debriefing.
>
> ***
> # More qualitative detail on case studies
> > W4: include more concrete examples and interview questions
>
> We agree. We already reference case examples throughout the paper for most subsections, eg: the agent architecture of an insurance agent (C1) in S5.4. In the revision, we will add more details for the qualitative examples where space and confidentiality permit, and otherwise expand the appendix and point to them more clearly.
>
> ***
> # Clarity on sample size
> > W5: clarify the production sample size
>
> We agree and will make this more explicit. The paper uses **306 total respondents** for broad survey context, but the main production analysis focuses on **86 deployed systems** plus **20 in-depth case studies**. We will revise the abstract and methodology so this distinction is immediately clear.
>
> We will also clarify that these are not small systems. Many of the deployed systems serve real users at meaningful scale, ranging from **tens to millions of users** (Fig. 14C, Appendix B). In this setting, access itself is a major bottleneck, so a corpus of 86 deployed systems and 20 in-depth cases is meaningful.
>
> ***
> # Reliability failure modes
> > W6, Q2: what types of reliability failures affect practitioners most heavily?
>
> We conducted an additional pass over the interview data and will add these patterns to the main paper. Three recurring reliability challenges are: **incomplete evaluation coverage**, which forces teams to rely on expert review while building task-specific test sets; **correctness failures in complex or multimodal workflows**, which often trigger human verification; and **legacy system, security, and compliance constraints**, which limit functionality or narrow deployment scope. We will surface these finer-grained categories in the main text.
>
> ***
> Thank you again for the thoughtful review. Your comments help us clarify scope, strengthen the limitations discussion, and add more detailed reliability analysis. We hope these clarifications and additional analyses address your concerns and make the contribution clearer. If so, we would be grateful if you could reflect that in your updated assessment. Looking forward to hearing from you.
>
> Best,
>
> The Authors

---

> > ### Author Rebuttal · Reviewer_z4SD · 2026-04-06
> >
> > Thank you for your considerate response! I think you've addressed my most substantive points, I will keep my score at a 5.

---

### Decision · Program_Chairs · 2026-04-30

**Decision:**

Accept (spotlight)

**Comment:**

This paper presents CAP, the first systematic empirical study of LLM agents in production. It combines 20 in-depth case studies with a survey of 306 practitioners across 26 domains, with core analysis on 86 deployed or pilot systems. Key findings are that production agents are intentionally simple and human-centered: 68% run at most 10 steps before human intervention, 70% use prompting of off-the-shelf models rather than weight tuning, and 74% rely primarily on human evaluation. Reliability emerges as the dominant deployment challenge.

All three reviewers recommend Accept after rebuttal.

Strengths: timely and important topic, novel first-hand data with survey instruments released, actionable findings that contrast with common research assumptions, clear structure and presentation.

Weaknesses: the abstract initially overstates the production sample size, geographic concentration in the Americas, data collection during a fast-moving period (April-November 2025), limited detail on grounded-theory coding in the main text, and reliability failure modes originally relegated to the appendix. Some findings are confirmatory.